# LoFT: L̲ow-Rank Adaptation That Behaves Like F̲ull Fine-T̲uning

**Nurbek Tastan**[1]    **Stefanos Laskaridis**[2]*   **Martin Takáč**[1]
**Karthik Nandakumar**[1,3]    **Samuel Horváth**[1]

[1]Mohamed bin Zayed University of Artificial Intelligence (MBZUAI), UAE
[2]Amazon Science, UK
[3]Michigan State University, USA
`{nurbek.tastan, martin.takac, samuel.horvath}@mbzuai.ac.ae`
`mail@stefanos.cc, nandakum@msu.edu`

## Abstract

Large pre-trained models are commonly adapted to downstream tasks using parameter-efficient fine-tuning methods such as Low-Rank Adaptation (LoRA), which injects small trainable low-rank matrices instead of updating all weights. While LoRA dramatically reduces trainable parameters with little overhead, it can still underperform full fine-tuning in accuracy and often converges more slowly. We introduce LoFT, a novel low-rank adaptation method that behaves like full fine-tuning by aligning the optimizer's internal dynamics with those of updating all model weights. LoFT not only learns weight updates in a low-rank subspace (like LoRA) but also properly projects the optimizer's first and second moments (Adam's momentum and variance) into the same subspace, mirroring full-model updates. By aligning the low-rank update itself with the full update, LoFT eliminates the need for tuning extra hyperparameters, e.g., the LoRA scaling factor $\alpha$. Empirically, this approach substantially narrows the performance gap between adapter-based tuning and full fine-tuning and consistently outperforms standard LoRA-style methods, all without increasing inference cost. The code is available at https://github.com/tnurbek/loft.

## 1 Introduction

Fine-tuning large-scale pre-trained models for specific tasks has become a standard paradigm in natural language processing and other domains. However, as model sizes grow into the billions of parameters, full fine-tuning (i.e., updating every weight) becomes computationally expensive and impractical, especially in multi-task (Chronopoulou et al., 2023) or multi-user (Yi et al., 2023) settings. Parameter-efficient fine-tuning (PEFT) techniques address this challenge by updating only a small subset of parameters while reusing the vast majority of pre-trained weights. Among these, Low-Rank Adaptation (LoRA) has emerged as a popular and effective solution. LoRA freezes the original weights and injects trainable low-rank matrices into selected layers, substantially reducing the number of learnable parameters. Remarkably, LoRA often matches – and sometimes can exceed – the performance of full fine-tuning on certain benchmarks, all while incurring minimal runtime overhead and no additional inference latency. This makes it an attractive alternative to other methods like sequential adapters (Houlsby et al., 2019; Pfeiffer et al., 2021), which typically introduce new layers and increased latency. Despite its success, LoRA and similar low-rank approaches still fall short of full fine-tuning in some settings. Empirical studies have reported a persistent performance gap and slower convergence rates compared to full-model updates (Biderman et al., 2024; Wang et al., 2024). These gaps indicate that the optimization dynamics of LoRA differ in important ways from those of full fine-tuning. Recent work (Liu et al., 2024; Wang et al., 2025) has attempted to close this gap by focusing on more accurate gradient approximations within the low-rank subspace. This is motivated by the observation that LoRA's updates can omit or misestimate important directions in the full gradient, leading to suboptimal solutions. In this work, we demonstrate that this is only part of the story: optimizer state misalignment – specifically in the first and second moments used by

---

*Work done independently of Amazon.

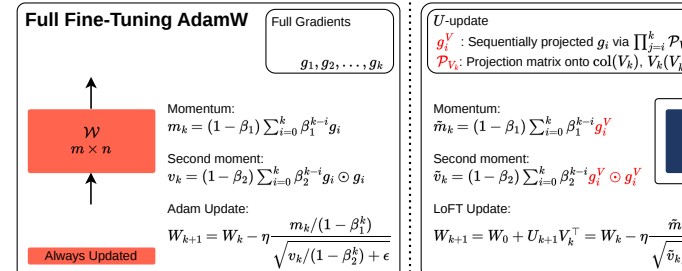

Figure 1: LoFT visualization. LoFT can be interpreted as the tightest approximation to full fine-tuning under the constraint that each update lies in the subspace defined by $V$ (when updating $U$). The LoFT-AdamW update consists of a momentum and second-moment estimate constructed using projected gradients. The final update is then projected back onto the subspace of $V$ to respect the low-rank constraint. When $V$ is the updated component instead of $U$, the roles of $U$ and $V$ are simply exchanged, and the update is applied to $W^\top$ instead of $W$.

AdamW (Loshchilov & Hutter, 2019), the de facto optimizer in large-scale training – also plays a critical role. When these internal statistics are not properly aligned with the low-rank constraint, it undermines the effectiveness of the adaptation.

Finally, a practical complication in standard LoRA is the introduction of a scaling hyperparameter, $\alpha$, often normalized by the rank. This scaling factor modulates the contribution of the low-rank update and must be carefully tuned. Improper settings can lead to poor performance or even divergence by overpowering the backbone model (Lee et al., 2025; Malinovsky et al., 2024). Altogether, these challenges – i.e., the gradient and optimizer state misalignment, as well as the additional hyperparameter sensitivity – limit LoRA's ability to fully replicate the robustness and effectiveness of unconstrained full fine-tuning.

Our main contributions are summarized as follows:

- We identify that not only gradients but also optimizer states (i.e., first and second moments) suffer from misalignment when approximating full fine-tuning with low-rank updates.
- We propose Low rank adaptation that mimics Full fine-Tuning (LoFT), a novel LoRA-based optimizer that addresses these issues by closely approximating full fine-tuning across all optimization dimensions. LoFT consists of five core components: gradient scaling, alternating updates, optimizer state calibration, construction of a projected full fine-tuning update followed by low-rank projection, and projected full fine-tuning-aware clipping.
- To the best of our knowledge, LoFT is the first low-rank adaptation method that exactly reduces to AdamW (Loshchilov & Hutter, 2019) in the full-rank limit.
- We conduct extensive experiments on both synthetic and real-world tasks across multiple modalities, demonstrating the effectiveness and generality of LoFT.

## 2 METHOD

We focus on the standard fine-tuning setup, where a pre-trained model is adapted to a downstream task. In full fine-tuning, each weight matrix $W$ is updated by a full-rank increment $\Delta W$. To reduce computational cost, LoRA proposes a low-rank reparameterization

$$W = W_0 + \Delta W = W_0 + UV^\top,$$

where $W \in \mathbb{R}^{m \times n}$, $U \in \mathbb{R}^{m \times r}$, $V \in \mathbb{R}^{n \times r}$, and $r \ll \min\{m, n\}$. Only $U$ and $V$ are trainable, reducing the gradient and optimizer state footprint to $\mathcal{O}((m + n)r)$ compared to $\mathcal{O}(mn)$ in full fine-tuning. LoRA typically introduces a scaling factor $\alpha > 0$ to modulate the magnitude of the low-rank update. However, in our study, we set $\alpha = 1$ and attribute the need for this hyperparameter to a misalignment between LoRA and full fine-tuning, which we address in the subsequent sections.

### 2.1 GRADIENT DESCENT FOR FULL FINE-TUNING VS. LORA

Let $f(W) : \mathbb{R}^{m \times n} \to \mathbb{R}$ denote a scalar loss function, with $W$ representing the parameters of a single linear layer. In standard full fine-tuning with gradient descent, the one-step update is

$$W^+ = W - \eta \nabla_W f(W), \tag{1}$$

Table 1: The six core building blocks of LoFT for aligning low-rank adaptation with full fine-tuning.

| Component | Purpose |
|---|---|
| Alternating Updates (1) | *Eliminate second-order cross terms from LoRA dynamics.* |
| Gradient Scaling (2) | *Ensure scale-invariance of low-rank updates.* |
| Optim. States Calibration (3, 4) | *Align moments estimates across changing low-rank spaces.* |
| Projected Full Update (5) | *Reconstruct the full-model update and project it onto the low-rank subspace.* |
| Gradient Clipping (6) | *Match full fine-tuning clipping behavior.* |

where $\eta > 0$ is the learning rate, and $\nabla_W f(W)$ is the gradient of the loss with respect to $W$. With LoRA parametrization, the one-step update becomes

$$W^+ = W_0 + U^+(V^+)^\top = W_0 + (U - \eta \nabla_U f(W))(V - \eta \nabla_V f(W))^\top. \tag{2}$$

Applying the chain rule yields

$$\nabla_U f(W) = \nabla_W f(W)V, \qquad \nabla_V f(W) = \nabla_W f(W)^\top U.$$

Substituting these into (2) gives

$$W^+ = W - \eta \left( \nabla_W f(W)VV^\top + UU^\top \nabla_W f(W) \right) + \eta^2 \nabla_W f(W)VU^\top \nabla_W f(W). \tag{3}$$

Equation (3) highlights the first discrepancy between LoRA and full fine-tuning: the additional $\eta^2$ term, which depends quadratically on the gradient. While seemingly small, this term can materially affect convergence, as we show later in a controlled experiment. A straightforward way to eliminate this term is through alternating updates.

> **Building Block 1: Alternating Updates**
>
> Do not update $U$ and $V$ simultaneously, but perform alternating updates.

Without loss of generality, assuming we update only $U$, the resulting update to $W$ becomes

$$W^+ = W - \eta \nabla_W f(W)VV^\top. \tag{4}$$

However, this update suffers from a scale ambiguity: for any $c \neq 0$, $UV^\top = (cU)(V/c)^\top$, but the update scales differently with $c$. To resolve this, observe that the update direction always lies in the column space of $V$, allowing us to scale the update using an $r \times r$ matrix[1] $(V^\top V)^{-1}$

$$W^+ = W - \eta \nabla_W f(W)V(V^\top V)^{-1}V^\top = W - \eta \nabla_W f(W)\mathcal{P}_V, \tag{5}$$

where $\mathcal{P}_V = V(V^\top V)^{-1}V^\top$ is the projection matrix onto the column space of $V$. This ensures the update is the closest low-rank approximation to $\nabla_W f(W)$ under the given subspace. The associated computational cost is $\mathcal{O}(nr^2 + r^3)$. This update defines our second building block.

> **Building Block 2: Use Scaled Gradients**
>
> $$\tilde{\nabla}_U f(W) = \nabla_U f(W)(V^\top V)^{-1}, \qquad \tilde{\nabla}_V f(W) = \nabla_V f(W)(U^\top U)^{-1}.$$

We are not the first to suggest this; Zhang & Pilanci (2024) derived a similar result from the perspective of Riemannian optimization.

## 2.2 FIRST MOMENT MISALIGNMENT

In practice, gradients are often estimated using momentum. Specifically, the first moment $m_k$ is computed as $m_k = \beta_1 m_{k-1} + (1 - \beta_1)g_k$, where $\beta_1 \in [0, 1)$ is the momentum coefficient and $g_k$ is

---

[1] We assume $V$ is of full rank. If not, we can use the pseudo-inverse.

the stochastic gradient, and the subscript denotes iteration counter. For full fine-tuning, the resulting momentum update is

$$m_k^W = (1 - \beta_1) \sum_{i=0}^{k} \beta_1^{k-i} \nabla_W f(W_i). \tag{6}$$

When updating $U$ under the LoRA parameterization, the effect on $W$ becomes

$$m_k^U V^\top = (1 - \beta_1) \sum_{i=0}^{k} \beta_1^{k-i} \tilde{\nabla}_U f(W_i) V^\top = (1 - \beta_1) \sum_{i=0}^{k} \beta_1^{k-i} \nabla_W f(W_i) V_i \left(V_i^\top V_i\right)^{-1} V_k^\top,$$

which does not represent a proper projection due to the mismatch between $V_i$ and $V_k$. To address this, we introduce a recalibration step

$$m_k^U = \beta_1 m_{k-1}^U C_k^V + (1 - \beta_1) \tilde{\nabla}_U f(W_k), \tag{7}$$

where $C_k^V \stackrel{\text{def}}{=} (V_{k-1}^\top V_k)(V_k^\top V_k)^{-1}$ is a calibration matrix. Substituting this back gives

$$\tilde{m}_k^U = m_k^U V_k^\top = (1 - \beta_1) \sum_{i=0}^{k} \beta_1^{k-i} \nabla_W f(W_i) \prod_{j=i}^{k} \mathcal{P}_{V_j} = (1 - \beta_1) \sum_{i=0}^{k} \beta_1^{k-i} g_i^V, \tag{8}$$

where $\prod_{j=i}^{k} \mathcal{P}_{V_j}$ is the sequential projection. Let $g_i^V \stackrel{\text{def}}{=} \nabla_W f(W_i) \prod_{j=i}^{k} \mathcal{P}_{V_j}$. This expression provides the tightest possible estimate (in $\ell_2$ distance) of the momentum under the constraints of the evolving low-rank subspaces defined by $V_i$'s. Storing previous iterates $\{V_{k-1}, U_{k-1}\}$ incurs an additional memory cost of $\mathcal{O}((m + n)r)$.

---

**Building Block 3: Recalibrate Momentum**

$$m_k^U = \beta_1 m_{k-1}^U C_k^V + (1 - \beta_1) \tilde{\nabla}_U f(W_i),$$
$$m_k^V = \beta_1 m_{k-1}^V C_k^U + (1 - \beta_1) \tilde{\nabla}_V f(W_i).$$

---

## 2.3 SECOND MOMENT MISALIGNMENT

Analogically, for Adam-style updates, the ideal update to $W$ when $U$ is being updated, given subspace constraints, would be

$$\frac{\tilde{m}_k^U / (1 - \beta_1^k)}{\sqrt{\tilde{v}_k^U / (1 - \beta_2^k)} + \varepsilon} \mathcal{P}_{V_k}, \text{ s.t. } \tilde{m}_k^U = (1 - \beta_1) \sum_{i=0}^{k} \beta_1^{k-i} g_i^V, \quad \tilde{v}_k^U = (1 - \beta_2) \sum_{i=0}^{k} \beta_2^{k-i} g_i^V \odot g_i^V, \tag{9}$$

where $\tilde{m}_k^U$ is as defined in (8), and $\tilde{v}_k^U = (1 - \beta_2) \sum_{i=0}^{k} \beta_2^{k-i} g_i^V \odot g_i^V$ is the second moment estimate. The symbol $\odot$ denotes element-wise multiplication. Note that this update is constructed to lie in the subspace defined by $V_k$, since this is a necessary constraint due to the update rule; see (4). To compute $\tilde{v}_k^U$ efficiently, we use the following identities from Slyusar (1999)

$$(A \bullet B)(C \otimes D) = (AC) \bullet (BD), \qquad (AB) \odot (CD) = (A \bullet C)(B * D) \tag{10}$$

where $\otimes$ is the Kronecker product, $\bullet$ is the transposed Khatri-Rao product, and $*$ is the standard Khatri-Rao product. We define the calibrated second-moment accumulator as

$$p_k^U = \beta_2 p_{k-1}^U (C_k^V \otimes C_k^V) + (1 - \beta_2)(\tilde{\nabla}_U f(W_k) \bullet \tilde{\nabla}_U f(W_k)), \tag{11}$$

where $p_k^U$ is a matrix of size $nr \times r$ that stores the cross-terms necessary to reconstruct the second moment after transformation. The associated memory overhead is $\mathcal{O}((m + n)r^2)$, which is the **main limitation** of our approach. For this reason, maintaining a small rank $r$ is crucial for memory efficiency. In practice, this constraint is acceptable as long as $r \leq \sqrt{\min\{m, n\}}$, which we find to be both reasonable and sufficient for capturing effective low-rank updates. In the experiment, LoFT leads to the memory increase of up to $25.65\%$ compared to LoRA (Hu et al., 2022), but **improves or matches** the memory of more performant DoRA (Liu et al., 2024). Furthermore, we observe

that omitting second-moment calibration limits memory increase to less than $6\%$ relative to LoRA and only incurs marginal performance degradation ($\sim 0.1\%$). Details are provided in Appendix E.2. Finally, in Appendix D.1, we introduce a version of LoFT based on the Muon (Jordan et al., 2024) optimizer that does not exhibit this issue, as all optimizer states are linear functions of gradients.

---

**Building Block 4: Second Moment Alignment**

Use cross-terms for second moment accumulation to enable second moment recalibration

$$p_k^U = \beta_2 p_{k-1}^U (C_k^V \otimes C_k^V) + (1 - \beta_2)(\tilde{\nabla}_U f(W_i) \bullet \tilde{\nabla}_U f(W_i)),$$
$$p_k^V = \beta_2 p_{k-1}^V (C_k^U \otimes C_k^U) + (1 - \beta_2)(\tilde{\nabla}_V f(W_i) \bullet \tilde{\nabla}_V f(W_i)). \qquad (12)$$

---

Using $p_k^U$, we compute $\tilde{v}_k^U = p_k^U (V_k * V_k)$ and apply the following update.

---

**Building Block 5: Reconstruct Full Update Followed by Projection**

For the Adam version of LoFT, update $U$ and $V$ as

$$U_{k+1} = U_k - \eta_k \frac{m_k^U V_k^\top / (1 - \beta_1^k)}{\sqrt{p_k^U (V_k * V_k) / (1 - \beta_2^k)} + \varepsilon} V_k (V_k^\top V_k)^{-1},$$
$$V_{k+1} = V_k - \eta_k \frac{m_k^V U_k^\top / (1 - \beta_1^k)}{\sqrt{p_k^V (U_k * U_k) / (1 - \beta_2^k)} + \varepsilon} U_k (U_k^\top U_k)^{-1}. \qquad (13)$$

---

### 2.4 GRADIENT CLIPPING AND WEIGHT DECAY

We apply no special modifications to weight decay. Since only one of $U$ or $V$ is updated at a time, the effect of standard weight decay correctly reduces the low-rank update as $UV^\top \rightarrow (1 - \lambda\eta)UV^\top$. The full AdamW-LoFT algorithm is provided in the appendix. *With all six building blocks described above, LoFT-AdamW recovers full fine-tuning when $r = \max\{m, n\}$ and $U_k, V_k$ are full-rank. To our knowledge, LoFT is the first low-rank adaptation method that provably recovers full fine-tuning.*

---

**Building Block 6: Gradient Clipping**

To approximate full fine-tuning during gradient clipping, when updating $U$, we use $\tilde{\nabla}_U f(W)V^\top = \nabla_W f(W)\mathcal{P}_W$ as the effective gradient for the corresponding layer $W$.

---

### 2.5 SIMULATED EXPERIMENT

In the previous remark, we argued that for full-rank adaptation, LoFT recovers full fine-tuning. We now demonstrate that if the target solution is low-rank, LoFT matches the performance of full fine-tuning if the correct rank is selected. We consider the optimization problem $f(W) = \|W - A\|_F^2$, where $A$ is a randomly generated matrix with $\text{rank}(A) = r$. We compare LoFT, LoRA, and full fine-tuning using the AdamW optimizer. To demonstrate how LoFT can efficiently approximate full fine-tuning, the step size is tuned for full fine-tuning and reused for all baselines. We initialize $W = 0$, and for LoFT we follow the standard LoRA initialization (Hu et al., 2022), which also yields $UV^\top = 0$ initially. We set

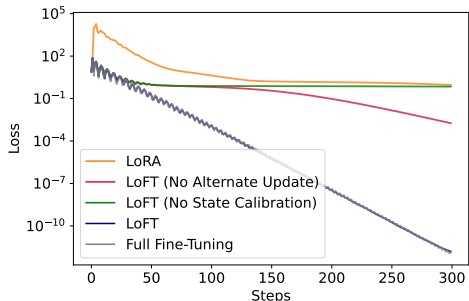

Figure 2: Comparison of LoRA, LoFT, and Full Fine-tuning with Adam on $f(W) = \|W - A\|_F^2$.

$m = 1024$, $n = 512$, and $r = 8$. In addition to LoFT and LoRA, we also include ablated variants of LoFT to highlight the importance of its design components: one without alternating updates, and one without optimizer state calibration. As shown in Figure 2, LoFT closely matches the performance of full fine-tuning. In contrast, omitting any of its core components leads to significantly slower convergence and worse final performance, confirming the necessity of the full LoFT design.

Table 2: Performance comparison of parameter-efficient fine-tuning methods, LoRA, DoRA, and our method LoFT, on a suite of commonsense reasoning benchmarks using LLaMA-7B, LLaMA2-7B, and LLaMA3-8B models. The table reports accuracy scores across multiple tasks with average performance shown in the final column. $r$ denotes the rank used in the respective adaptation method. **Bold** and underlined scores highlight the best and second-best performance per task, respectively.

| Model | Method | BoolQ | PIQA | SIQA | HS | WG | ARC-C | ARC-E | OBQA | avg. |
|-------|--------|-------|------|------|------|------|-------|-------|------|------|
| LLaMA-7B | LoRA$_{r=16}$ | 65.38 | 76.71 | 75.69 | 79.81 | 68.03 | **65.27** | 80.30 | 77.40 | 73.57 |
| | DoRA$_{r=16}$ | 54.13 | 73.94 | **79.38** | 58.01 | **79.40** | 64.68 | 79.76 | **79.60** | 71.11 |
| | LoFT$_{r=16}$ | **68.62** | **82.80** | 78.27 | **82.69** | 73.32 | 64.30 | 80.26 | 78.40 | **76.08** |
| | LoFT$_{r=4}$ | 67.34 | 80.96 | 76.20 | 80.50 | 76.40 | 63.62 | 79.21 | 75.40 | 74.95 |
| | LoFT$_{r=2}$ | 68.03 | 79.16 | 75.84 | 78.86 | 76.24 | 64.51 | 78.03 | 71.00 | 73.96 |
| | LoFT$_{r=1}$ | 67.09 | 78.35 | 74.46 | 76.14 | 74.82 | 58.87 | 76.85 | 70.80 | 72.17 |
| LLaMA2-7B | LoRA$_{r=16}$ | 50.09 | 59.03 | 76.41 | 65.45 | 77.51 | 64.68 | 79.12 | 77.20 | 68.69 |
| | DoRA$_{r=16}$ | **71.93** | 82.92 | 79.22 | 88.90 | **83.03** | 66.98 | 82.70 | **82.00** | 79.71 |
| | LoFT$_{r=16}$ | 71.80 | **83.51** | 79.02 | **90.59** | 82.72 | **70.65** | 84.43 | 81.00 | **80.46** |
| | LoFT$_{r=4}$ | 70.49 | 81.94 | **79.80** | 88.85 | 81.37 | 69.11 | **84.88** | 79.80 | 79.53 |
| | LoFT$_{r=2}$ | 70.55 | 81.18 | 77.74 | 83.01 | 79.01 | 66.72 | 82.83 | 78.80 | 77.48 |
| | LoFT$_{r=1}$ | 68.69 | 80.58 | 76.36 | 72.95 | 76.80 | 64.08 | 82.37 | 77.20 | 74.88 |
| LLaMA3-8B | LoRA$_{r=16}$ | 74.46 | 88.14 | **81.37** | 94.81 | 85.08 | 80.72 | 89.18 | 86.00 | 84.97 |
| | DoRA$_{r=16}$ | 74.56 | 88.52 | 80.09 | 95.17 | **86.74** | 79.78 | 90.19 | 84.60 | 84.96 |
| | LoFT$_{r=16}$ | **75.63** | **88.85** | 80.35 | **95.64** | 86.11 | **80.89** | **91.16** | 86.40 | **85.63** |
| | LoFT$_{r=4}$ | 74.53 | 88.52 | 80.04 | 95.45 | 85.32 | 78.92 | 89.73 | 84.20 | 84.59 |
| | LoFT$_{r=2}$ | 73.76 | 87.11 | 79.84 | 94.72 | 84.29 | 79.61 | 89.98 | 84.60 | 84.24 |
| | LoFT$_{r=1}$ | 69.33 | 87.49 | 79.27 | 93.79 | 84.06 | 76.11 | 87.12 | 82.20 | 82.42 |

## 3 EXPERIMENTS

We conduct extensive experiments across both language and vision domains to evaluate the effectiveness of our method. Our primary baselines include LoRA (Hu et al., 2022), DoRA (Liu et al., 2024), and full fine-tuning, and we apply these methods to a range of model backbones: LLaMA-7B (Touvron et al., 2023a), LLaMA2-7B (Touvron et al., 2023b), LLaMA3-8B (Grattafiori et al., 2024), LLaMA3.1-70B (Grattafiori et al., 2024), and ViT-Base (Wu et al., 2020). The evaluation spans two major fronts: (i) commonsense reasoning tasks in the language domain, and (ii) image classification tasks involving highly imbalanced and domain-specific datasets, including several medical imaging datasets and DomainNet. We focus on LoRA and DoRA as our primary baselines since they are the most widely adopted and directly comparable PEFT methods, while results with additional baselines (namely full finetuning, rsLoRA (Kalajdzievski, 2023), AdaLoRA (Zhang et al., 2023), LoRA-Pro (Wang et al., 2025), LoRA-GA (Wang et al., 2024), LoRA$^+$ (Hayou et al., 2024) ) are provided in Appendix E.4.

In addition to the typical low-rank configuration (e.g., $\text{rank} \geq 4$), we explore extremely constrained settings by reducing the rank to as low as 1, demonstrating the robustness of our method under stringent parameter budgets. This allows us to highlight not just absolute performance but also the parameter efficiency and scalability of our approach relative to existing baselines. Further implementation and dataset details are provided in Appendix C. For additional baselines, LoFT derivatives (LoFT (simple), which removes second-moment calibration to reduce memory and latency overhead, and quantized LoFT), as well as ablations, memory footprint, and latency analysis, please refer to Appendix E. Large-scale scaling results on LLaMA-3.1-70B are provided in Appendix F.3.

### 3.1 COMMONSENSE REASONING

**Setup.** To evaluate the efficacy of LoFT in the language domain, we conduct experiments on a suite of commonsense reasoning benchmarks, including BoolQ, PIQA, SIQA, HellaSwag (HS), Winogrande (WG), ARC-Challenge (ARC-C), ARC-Easy (ARC-E), and OpenBookQA (OBQA). We fine-tune three prominent large language models, LLaMA-7B (Touvron et al., 2023a), LLaMA2-7B (Touvron et al., 2023b), and LLaMA3-8B (Grattafiori et al., 2024), using parameter-efficient methods: LoRA, DoRA, and our proposed LoFT, each evaluated at multiple rank settings, notably including very low ranks (e.g., $1, 2, 4$). Following the setting of Hu et al. (2023), we combine the training sets from all eight benchmarks into a single unified training dataset, and then conduct evaluation separately on each task's official test set. This unified training strategy enables more stable fine-tuning and fairer comparisons across tasks and adaptation methods.

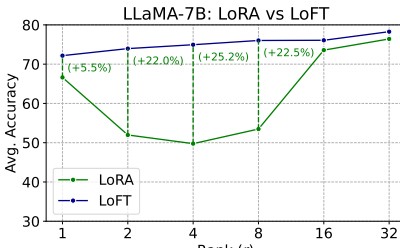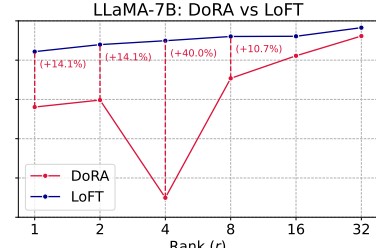

Figure 3: Rank-wise comparison of LoFT against LoRA (left) and DoRA (right) on LLaMA-7B across commonsense reasoning tasks. LoFT maintains significantly higher accuracy, especially at low ranks. Percentage gains denote improvement of LoFT over the respective baseline at each rank.

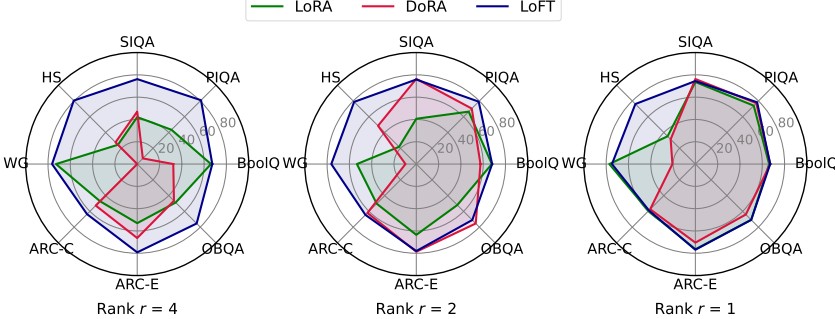

Figure 4: Task-wise performance comparison across LoRA (green), DoRA (red), and LoFT (blue) at lower ranks ($r = \{4, 2, 1\}$) on LLaMA-7B. LoFT maintains high performance across all tasks, even under extreme compression, unlike baselines that degrade sharply on several benchmarks.

**Overall Performance Results.** As shown in Table 2, LoFT consistently achieves superior performance across all model scales and rank configurations. For LLaMA-7B, LoFT at rank 16 achieves the highest average accuracy of 76.08%, outperforming both LoRA (73.57%) and DoRA (71.11%) by notable margins. Even at lower ranks, LoFT maintains strong performance, only a 1.1% drop at rank 4 and 3.9% at rank 1, demonstrating its robustness in extremely low-rank regimes. The trend continues for LLaMA2-7B, where LoFT at rank 16 reaches an average accuracy of 80.46%, surpassing LoRA by 11.7% and slightly edging out DoRA. Remarkably, LoFT remains highly competitive down to rank 1, scoring 74.88%, which still outperforms LoRA by a significant margin. For the largest model, LLaMA3-8B, LoFT achieves the highest average accuracy of 85.63% at rank 16. The gains over LoRA and DoRA are less dramatic, but LoFT's performance remains consistently on top. Importantly, the drop-off in performance with decreasing rank is significantly more graceful for LoFT.

**Rank-Wise Comparison.** To better illustrate LoFT's robustness and performance scalability, we present a rank-wise comparison in Figure 3. The left panel compares LoFT against LoRA, and the right panel compares it against DoRA, both on LLaMA-7B. We observe that LoFT consistently outperforms both baselines across all rank settings, but the gap becomes especially pronounced at low ranks. Notably, at rank 4, LoFT surpasses DoRA by an impressive +40% and LoRA by +25%, highlighting LoFT's extreme efficiency in constrained settings.

Interestingly, while LoRA and DoRA both suffer steep accuracy drops at lower ranks, LoFT exhibits a much flatter accuracy curve, showing that it retains high performance even with minimal trainable parameters. This makes LoFT particularly appealing for low-resource deployment scenarios.

These results validate two important properties of our method: *(i) LoFT matches or exceeds the performance of existing PEFT methods even at high capacity ($r = 16$)*, and *(ii) it remains highly effective at extremely low ranks*, highlighting its efficiency and applicability in constrained settings. Overall, LoFT achieves the best balance between accuracy and parameter count across diverse commonsense reasoning tasks, while using the same number of parameters as LoRA.

**Task-Specific Analysis at Low Ranks.** To further analyze performance under parameter-constrained settings, we examine how LoRA, DoRA, and LoFT behave across individual tasks at lower ranks $r = \{4, 2, 1\}$ using LLaMA-7B. Figure 4 shows radar plots for all eight commonsense reasoning benchmarks at each of these low ranks. These visualizations reveal that while LoRA and DoRA suffer

Table 3: Comparison of parameter-efficient fine-tuning methods on image classification benchmarks using the ViT-Base model. We evaluate full fine-tuning (Full FT), LoRA, DoRA, and our proposed method, LoFT, across four datasets: ISIC2019, HAM10000, Diabetic Retinopathy, and DomainNet. Accuracy (mean ± standard deviation) is reported for each setting.

| Model | Method | ISIC2019 | HAM10000 | Diabetic Retinopathy | DomainNet | avg. |
|---|---|---|---|---|---|---|
| ViT-Base | Full FT | $80.69 \pm 0.18$ | $\textbf{93.22} \pm 0.64$ | $56.07 \pm 0.23$ | $\textbf{73.46} \pm 1.20$ | $\underline{75.86}$ |
| | LoRA$_{r=16}$ | $\underline{81.02} \pm 1.10$ | $91.56 \pm 0.66$ | $57.87 \pm 0.43$ | $71.39 \pm 0.10$ | $75.46$ |
| | DoRA$_{r=16}$ | $80.35 \pm 0.17$ | $90.78 \pm 0.81$ | $57.66 \pm 0.56$ | $70.18 \pm 2.02$ | $74.74$ |
| | LoFT$_{r=16}$ | $\textbf{81.06} \pm 0.13$ | $\underline{93.13} \pm 0.28$ | $\textbf{58.33} \pm 0.19$ | $\underline{71.97} \pm 0.16$ | $\textbf{76.12}$ |
| | LoFT$_{r=8}$ | $80.36 \pm 0.21$ | $91.78 \pm 0.68$ | $\underline{57.89} \pm 0.48$ | $70.11 \pm 0.77$ | $75.04$ |
| | LoFT$_{r=4}$ | $79.31 \pm 0.36$ | $91.45 \pm 0.73$ | $56.98 \pm 0.27$ | $69.32 \pm 0.55$ | $74.27$ |

inconsistent and often sharp performance drops across tasks, LoFT maintains stable and competitive accuracy across the board.

In particular, DoRA shows substantial instability at ranks 4 and 2, with near zero scores on certain tasks such as WinoGrande, whereas LoRA suffers large dips on more complex tasks like HellaSwag and SIQA. In contrast, LoFT retains high task-wise accuracy, especially on harder benchmarks (e.g., HellaSwag, ARC-C), even at rank 1, demonstrating its robust generalization when adaptation budgets are extremely constrained. For a comprehensive view of the exact numerical breakdowns per task and rank, we refer readers to the appendix.

## 3.2 IMAGE CLASSIFICATION

To assess the generality of our approach beyond the language domain, we evaluate it on image classification tasks using the ViT-Base model (Wu et al., 2020) pretrained on ImageNet-21K (Deng et al., 2009). Vision models are known to be more sensitive to low-rank constraints, often requiring higher intrinsic ranks to preserve performance. Therefore, we restrict our analysis to ranks $r \geq 4$ and focus on whether LoFT can match or exceed strong baselines under these challenging constraints.

We conduct experiments on four diverse and challenging datasets: **ISIC2019** (Codella et al., 2019) and **HAM10000** (Tschandl et al., 2018): medical skin lesion classification datasets with long-tailed label distributions; **Diabetic Retinopathy** (Graham, 2015): a medical imaging dataset with ordinal severity levels, and **DomainNet** (Peng et al., 2019): a large-scale highly skewed benchmark.

We compare our method (LoFT) against full fine-tuning (Full FT), LoRA, and DoRA using a consistent configuration (rank $r = 16$ unless specified otherwise). For each dataset, we report the mean and standard deviation over three runs.

As shown on Table 3, LoFT at rank 16 achieves the highest average accuracy 76.12%, outperforming both LoRA (75.46%) and DoRA (74.74%), and even slightly surpassing full fine-tuning (75.86%). LoFT also achieves the top score on two of four individual datasets, including ISIC2019 and Diabetic Retinopathy. Notably, LoRA performs competitively on ISIC2019 but exhibits degraded performance on HAM10000 and DomainNet, suggesting it may struggle with skewed datasets. DoRA generally underperforms across datasets, indicating instability in visual

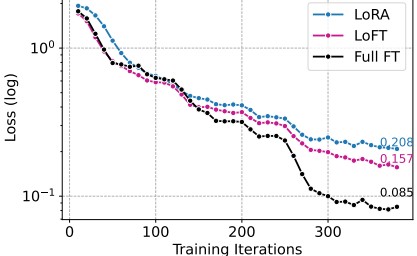

Figure 5: Training log. loss on HAM10000.

domains with skewed/out-of-domain datasets. In contrast, LoFT maintains strong performance, even when the rank is reduced to 8 and 4, with only a 2-point drop in average accuracy at rank 4, further reinforcing its resilience to low-rank degradation in vision tasks.

In addition to the final accuracy gains reported in Table 3, we also present the training dynamics on HAM10000 in Figure 5. Remarkably, LoFT's training loss curve closely overlaps with that of full fine-tuning from the very first iterations, indicating that our updates follow the same optimization trajectory as Full FT right from the start. In contrast, LoRA starts with a noticeably higher loss and converges more slowly, never fully matching Full FT's initial descent. This early alignment between LoFT and Full FT demonstrates that, despite updating far fewer parameters, LoFT preserves the model's capacity to adapt rapidly.

Throughout the remainder of training, LoFT maintains a small gap behind Full FT, which we attribute to the growing rank of the full fine-tuning solution, as explained by greedy low-rank learning theory (Li et al., 2021). Nevertheless, LoFT significantly outperforms LoRA across the full training trajectory. Interestingly, LoFT ultimately achieves better final performance than full fine-tuning, suggesting that Full FT may overfit, whereas LoFT benefits from implicit regularization due to the low-rank structure of its updates.

## 4 RELATED WORK

**Parameter-Efficient Fine-Tuning.** As aforementioned, the advent of Large Language Models has exploded the computational and memory requirements of running neural workloads, at training and inference time, thus limiting running such tasks to a few players. Towards this end, a significant amount of research has focused on efficient ways of fine-tuning LLMs for downstream tasks. Parameter Efficient Fine-Tuning (PEFT) collectively refers to techniques that only tune a small number of parameters towards the optimization objective. Such methods take various shapes, ranging from token-level (i.e., prompt-tuning) (Lester et al., 2021) and intermediate state parameters (i.e., prefix-tuning) (Li & Liang, 2021) to block-level parameters interspersed in the transformer block, either sequentially (Houlsby et al., 2019; Pfeiffer et al., 2021) or in parallel (He et al., 2022).

**Low-Rank Adaptation.** Closer to our method, LoRA (Hu et al., 2022) introduces low-rank adapters parallel to the attention and linear layers of the transformer block, which build upon the assumption that the changes in model weights during adaptation exhibit a low-rank structure and thus reparametrize updated weights as such. While seminal, LoRA often falls short of the full fine-tuning potential of the model. Subsequent work has tried to tackle this in various ways. Specifically, DoRA (Liu et al., 2024) decomposes the model weights into their directional and magnitude components and fine-tunes both, but only the former remains low-rank. Similar in nature is DeLoRA (Bini et al., 2025), decouples the direction and strength of low-rank weight updates via normalization and learnable scaling. On the contrary, Zhu et al. (2024) note the distinct function of $A$ and $B$ low-rank matrices and propose training only the latter for efficiency, while Hayou et al. (2024) adopts different learning rates for each matrix. LoRA-Pro (Wang et al., 2025) shows the equivalence of low-rank adaptation and low-rank gradient and enhances LoRA by minimizing the distance between the true gradient and the low-rank matrices A and B in closed form. Zhang & Pilanci (2024) introduce a Riemannian preconditioner to enhance the stability and efficiency of LoRA with SGD and AdamW optimizers across tasks. PiSSA (Meng et al., 2024), on the other hand, pinpoints the issue with the initialization of LoRA matrices and proposes SVD decomposition and freezing only the residual components of the weights. All of the above methods attempt to more faithfully approximate the gradients in the low-rank subspace and close the performance gap of LoRA with full fine-tuning. Contrary to prior work, our primary goal focuses on the optimization dynamics of low-rank models and aligning the optimizer state to full fine-tuning. By doing so, we are able to get state-of-the-art results without sacrificing accuracy or efficiency. Concurrently with our work, AltLoRA (Yu et al., 2025) proposes an alternating optimization scheme for LoRA that mitigates second-order coupling effects via alternating projections.[2] Both approaches share the central idea of alternating updates to alleviate coupling effects in low-rank parameterizations (i.e., alternating optimization constitutes the common core). However, they differ substantially in scope and objective. AltLoRA primarily focuses on improving gradient approximation within the low-rank subspace and establishes recovery of SGD with momentum. In contrast, our method extends beyond gradient alignment to match the full optimizer dynamics, aligning not only gradients but also all internal optimizer moments. This enables exact recovery of full AdamW dynamics in the low-rank regime and provable reduction to full fine-tuning in the full-rank limit. Moreover, we introduce projected full-update reconstruction and moment calibration matrices, and eliminate the need for the LoRA scaling hyperparameter. To our knowledge, our method is the first to establish transformation invariance under full AdamW dynamics, encompassing first- and second-moment alignment, gradient clipping, and weight decay.

**More efficient LoRA.** While low-rank adaptation significantly drops the computational and memory requirements of training large-scale LLMs, it still can require a significant amount of resources, especially in constrained edge or cross-device federated learning settings (Cho et al., 2024). Towards this end, several approaches further optimize low-rank adaptation to minimize the overhead. Specifically,

---

[2]AltLoRA and our work were developed independently and appeared on arXiv within days of each other.

VeRA (Kopiczko et al., 2024) proposes freezing shared random low-rank matrices and only training scaling vectors. LoRA-xs (Bałazy et al., 2024) freezes SVD-initialized low-rank matrices and only trains a small $r \times r$ matrix for adaptation. Last, LoRA-SB (Ponkshe et al., 2024) more carefully initializes the low-rank matrices to more faithfully approximate the full fine-tuning gradient directions during adaptation. Contrary to such approaches, LoFT can scale to truly low ranks by careful tuning of the optimization process, rather than altering the adaptation modeling.

## 5 CONCLUSION

In this work, we have presented LoFT, a low-rank adaptation framework that aligns the optimizer's internal dynamics to full fine-tuning by means of alternating LoRA updates, gradient projection and scaling, first and second moment calibration, and gradient clipping approximations. These mechanisms enable significant performance and efficiency gains with minimal loss in accuracy across different tasks and model sizes and pave the way for training even more efficiently for downstream tasks. Towards this end, we plan to explore the interplay between our LoFT and quantization to further boost efficiency and sustainability in training, as well as how it can be combined with noisy Differential Privacy updates, which can enable distributed private training at scale.

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

# Table of Contents

## A  LIMITATIONS

In this paper, we have proposed a technique for producing parameter-efficient fine-tuning via low-rank adaptation that behaves like full fine-tuning. While LoFT offers significant efficiency gains compared to full fine-tuning and DoRA, it has increased memory consumption compared to LoRA (due to storing previous iterates $V_{k-1}, U_{k-1}$), to the benefit of increased accuracy (see Section E.2). Having said that, even for the same memory footprint of LoRA, LoFT is able to achieve better downstream accuracy.

Lastly, we applied our technique on top of the AdamW optimizer, as it is the most widely used in LLM optimization. We leave applications to other optimizers, like Muon, for future work.

## B  THEORETICAL PROPERTIES OF LoFT FOR MATRIX FACTORIZATION

In Section 2.4, we argue that when $UV^\top$ is of full rank, then LoFT recovers full fine-tuning. Furthermore, for the matrix factorization problem, we showed that if the true solution is of low rank, then LoFT also empirically recovers full fine-tuning. In this section, we further extend these results. In particular, we focus on the matrix factorization problem

$$\min_{U \in \mathbb{R}^{m \times r}, V \in \mathbb{R}^{n \times r}} \left\{ f(U, V) \stackrel{\text{def}}{=} \frac{1}{2} \|UV^\top - A\|_F^2 \right\}. \tag{14}$$

Let $A = \tilde{U}\Sigma\tilde{V}^\top$ be the SVD decomposition of $A$. Then, by the Eckart-Young theorem, we have that every solution of (14) has the following form:

$$U^\star = \tilde{U}_r \Sigma_r Q,$$
$$V^\star = \tilde{V}_r \left(Q^{-1}\right)^T,$$

where $\tilde{U}_r, \Sigma_r, \tilde{V}_r$ contain the first $r$ singular vectors of $A$ and $Q \in \mathbb{R}^{r \times r}$ is a full rank matrix. In the next lemma, we show that if $U$ and $V$ start in the correct space, then LoFT applied to gradient descent with momentum recovers full fine-tuning with momentum.

**Lemma 1.** *Let $U_0 = \tilde{U}_r X_0$ and $V_0 = \tilde{V}_r Y_0$, where $X_0, Y_0 \in \mathbb{R}^{r \times r}$ are full rank matrices. Then, LoFT-GD with momentum applied to the matrix factorization problem exactly recovers GD with momentum applied to $f(W) = \frac{1}{2}\|W - A\|_F^2$ initialized at $W_0 = U_0 V_0^\top$.*

*Proof.* The gradient of $f(W)$ with respect to $W$ has the following form:

$$\nabla_W f(W_0) = W_0 - A = \tilde{U}_r \left(X_0 Y_0^\top - \Sigma_r\right) \tilde{V}_r^\top.$$

The left and right spaces correspond to $\tilde{U}_r$ and $\tilde{V}_r$, respectively. Using (5) and (8), we get

$$g_0^V = g_0^U = \nabla_W f(W_0) \text{ and } \tilde{m}_0^V = \tilde{m}_0^U = m_0 = \nabla_W f(W_0).$$

Since momentum is also the update, we have by induction that $\forall k \geq 0$, $U_k = \tilde{U}_r X_k$ and $V_k = \tilde{V}_r Y_k$, where $X_k, Y_k \in \mathbb{R}^{r \times r}$. Therefore,

$$g_k^V = g_k^U = \nabla_W f(W_k), \text{ and}$$

$$\tilde{m}_k^V = \tilde{m}_k^U = m_k = (1 - \beta_1) \sum_{i=0}^k \beta_1^{k-i} \nabla_W f(W_i).$$

$\square$

One interesting consequence of the above lemma is that if we apply LoFT with step size 1 with the initialization in the correct space, LoFT finds the optimal solution in a single step. Notice that without scaling, the smoothness constant of (14) with respect to both optimization variables can be unbounded, since

$$\|\nabla_U f(U_1, V) - \nabla_U f(U_2, V)\|_F = \|(U_1 V^T - A)V - (U_2 V^T - A)V\|_F$$
$$= \|(U_1 - U_2)V^\top V\|_F$$

can be unbounded as $\|V\|_F$ can be unbounded. In practice, we would need to restrict $\|U\|_F$ and $\|V\|_F$ to guarantee smoothness. On the other hand, LoFT scaled version of the gradient satisfies:

$$
\begin{aligned}
\|\tilde{\nabla}_U f(U_1, V) - \tilde{\nabla}_U f(U_2, V)\|_F &= \|(U_1 V^T - A)V(V^\top V)^{-1} - (U_2 V^T - A)V(V^\top V)^{-1}\|_F \\
&= \|(U_1 - U_2)V^\top V(V^\top V)^{-1}\|_F \\
&= 1\|U_1 - U_2\|_F.
\end{aligned}
$$

Therefore, LoFT gradients are smooth with the smoothness constant 1 without any restrictions. The above highlights another desirable property of LoFT introduced in the following lemma.

**Lemma 2.** *LoFT-GD with step size $\eta = 1$ applied to the matrix factorization* (14) *corresponds to the Alternating Least Squares algorithm.*

*Proof.* Without loss of generality, we assume $U$ is updated. Let $E_k = U_k V_k^\top - A$, then:

$$
\begin{aligned}
E_{k+1} = U_{k+1}V_k^\top - A &= \left(U_k - E_k V_k \left(V_k^\top V_k\right)^{-1}\right)V_k^\top - A \\
&= E_k - E_k V_k \left(V_k^\top V_k\right)^{-1} V_k^\top \\
&= E_k \left(I - \mathcal{P}_{V_k}\right).
\end{aligned}
$$

Therefore,

$$
f(U_{k+1}, V_k) = \frac{1}{2}\|E_{k+1}\|_F^2 = \frac{1}{2}\|E_k \left(I - \mathcal{P}_{V_k}\right)\|_F^2 = \min_{U \in \mathbb{R}^{m \times r}} \frac{1}{2}\|UV_k^\top - A\|_F^2 = \min_{U \in \mathbb{R}^{m \times r}} f(U, V_k).
$$

Analogically, we can derive

$$
f(U_k, V_{k+1}) = \min_{V \in \mathbb{R}^{n \times r}} f(U_k, V),
$$

which concludes the proof.

$\square$

# C    IMPLEMENTATION DETAILS

**Compute Information.**    All experiments reported in this paper were conducted using a single NVIDIA A100-SXM4-40GB GPU. This setup was used consistently across all experimental runs. Time of execution and memory usage varied slightly depending on the model configuration, but all runs were completed on a single-GPU setup. No additional or external compute (e.g., cloud services) was used during these experiments.

The implementation of LoFT used in our experiments can be found at: https://github.com/tnurbek/loft.

## C.1    DATASETS

**Commonsense Reasoning.**    To evaluate language models' reasoning capabilities, we use a curated commonsense reasoning benchmark COMMONSENSE170K (Hu et al., 2023) consisting of 170K diverse examples. These examples are drawn from multiple existing commonsense QA datasets and span a variety of tasks, including physical reasoning, social intuition, temporal understanding, and cause-effect inference.

**Image Classification.**    We conduct experiments on four diverse and challenging datasets to evaluate the generalization ability of our method in the image classification domain:

- **ISIC2019** (Codella et al., 2019) is a medical dataset composed of $25300$ training and $8238$ test dermoscopic images spanning eight skin lesion categories. It presents a long-tailed distribution, with the largest class heavily overrepresented relative to rare malignancies such as dermatofibroma or vascular lesions. The dataset is particularly challenging due to inter-class visual similarity and intra-class variability.

- **HAM10000** (Tschandl et al., 2018) contains $\{8.2K + 1.2K\}$ (training + test) high-resolution dermoscopic images categorized into seven skin lesion types. In includes lesions from diverse populations and acquisition sources. Similar to ISIC2019, this dataset suffers from severe class imbalance.

- **Diabetic Retinopathy** (Graham, 2015) consists of $\{115K + 14.2K\}$ (training + test) retinal fundus images annotated with ordinal labels representing five stages of diabetic retinopathy severity. The task involves predicting these severity levels from fundus scans.

- **DomainNet** (Peng et al., 2019) is a large-scale dataset designed for domain generalization and adaptation. It contains approximately $587000$ images from $345$ categories across six domains: real, clipart, infograph, painting, quickdraw, and sketch. Its substantial domain shift and high class diversity make it a valuable benchmark for testing superiority of the methods.

**Math Reasoning.**    To assess mathematical reasoning in large language models, we use the ORCA-MATH dataset (Mitra et al., 2024), a benchmark of 200K diverse math problems spanning arithmetic, algebra, geometry, calculus, and probability. Each problem requires multi-step reasoning and symbolic manipulation, making the dataset well-suited for evaluating fine-tuning strategies.

**Language Modeling.**    To evaluate language modeling and text generation under low-resource conditions, we use the WIKITEXT2 dataset (Merity et al., 2017), a widely adopted benchmark consisting of over 100K tokens from cleaned Wikipedia articles. The dataset preserves natural long-range dependencies by retaining full articles and punctuation, making it suitable for assessing perplexity and generalization in autoregressive models. We follow the original data split and preprocessing protocol established by Radford et al. (2019).

## C.2    HYPERPARAMETERS

We report training configurations for the main experiments: commonsense reasoning and image classification. For clarity and reproducibility, the full hyperparameter settings for each task are presented in tables below. Hyperparameters for the remaining tasks, including math reasoning, language modeling, and code generation, are detailed separately in Sections F.2, E.4, H, respectively.

Table 4: Hyperparameter configurations used for LoRA/DoRA (as in (Liu et al., 2024)) and our method, LoFT, across LLaMA model variants on commonsense reasoning tasks. Unlike prior method that tune the LoRA scaling factor $\alpha$, LoFT sets $\alpha = r$ consistently across all models without the need for tuning.

| Hyperparameter | LoRA/DoRA | | | LoFT | | |
|---|---|---|---|---|---|---|
| | LLaMA-7B | LLaMA2-7B | LLaMA3-8B | LLaMA-7B | LLaMA2-7B | LLaMA3-8B |
| **Rank** $r$ | | $r$ | | | $r$ | |
| **Alpha scaler** $\alpha$ | | $2 \times r$ | | | $r$ | |
| **Dropout** | | 0.05 | | | 0.05 | |
| **Optimizer** | | AdamW | | | AdamW | |
| **Learning rate** | $2 \times 10^{-4}$ | $3 \times 10^{-4}$ | $1 \times 10^{-4}$ | $2 \times 10^{-4}$ | $3 \times 10^{-4}$ | $1 \times 10^{-4}$ |
| **LR scheduler** | | Linear | | | Linear | |
| **Batch size** | | 16 | | | 16 | |
| **Micro-batch size** | | 16 | | | 16 | |
| **Warmup steps** | | 100 | | | 100 | |
| **Training epochs** | | 3 | | | 3 | |
| **Low-rank targets** | | $\mathrm{Q, K, V, Up, Down}$ | | | $\mathrm{Q, K, V, Up, Down}$ | |

Table 5: Training hyperparameters for ViT-B/16 across four image classification datasets. All methods (Full FT, LoRA, DoRA, and LoFT) are trained using the same configuration for fair comparison.

| Dataset | Rank $r$ | Batch Size | LR | Epochs | Target Modules | LoRA/DoRA $\alpha$ | LoFT $\alpha$ |
|---|---|---|---|---|---|---|---|
| ISIC2019 | $r$ | 64 | $5 \times 10^{-4}$ | 3 | $\mathrm{Q, K, V, Dense}$ | $2 \times r$ | $r$ |
| HAM10000 | $r$ | 64 | $5 \times 10^{-4}$ | 3 | $\mathrm{Q, K, V, Dense}$ | $2 \times r$ | $r$ |
| Retinopathy | $r$ | 64 | $5 \times 10^{-4}$ | 3 | $\mathrm{Q, K, V, Dense}$ | $2 \times r$ | $r$ |
| DomainNet | $r$ | 256 | $5 \times 10^{-4}$ | 3 | $\mathrm{Q, K, V, Dense}$ | $2 \times r$ | $r$ |

*Common settings:* Optimizer = AdamW, LR scheduler = Linear, Warmup ratio = 0.1, Dropout = 0.1, Micro-batch size = Batch size.

**Commonsense Reasoning.** We evaluate three generations of LLaMA family models, LLaMA-7B, LLaMA2-7B, and LLaMA3-8B, to test whether our proposed **LoFT** approach scales consistently across architectural updates. For each backbone, we compare against two strong parameter-efficient baselines, LoRA (Hu et al., 2022) and DoRA (Liu et al., 2024). For these experiments, we adopt the optimal hyperparameter settings reported in (Liu et al., 2024). We adopt the same learning rate, learning rate scheduler, warmup steps, batch size, and the same $\mathrm{Q, K, V, Up, Down}$ matrices for applying LoRA. The full configuration is summarized in Table 4.

**Image Classification.** We conduct image classification experiments using the ViT-B/16 model across four datasets: ISIC2019, HAM10000, Diabetic Retinopathy, and DomainNet. The input resolution is fixed to $224 \times 224$ pixels, and the patch size is set to 16. All methods, including full fine-tuning, LoRA, DoRA, and our proposed LoFT, share the same training configuration to ensure a fair comparison.

Specifically, we fix the learning rate to $5 \times 10^{-4}$ across all datasets. The batch size is set to 64 for medical datasets and increased to 256 for DomainNet due to its scale. All models are trained for 3 epochs using the AdamW optimizer, with a linear learning rate scheduler and a warmup ratio of 0.1. A dropout rate of 0.1 is applied, and low-rank methods target both the $\mathrm{Q, K, V}$ attention layers and the $\mathrm{Dense}$ layers. These hyperparameters are summarized in Table 5.

**Scaling Factor.** We clarify the role of the scaling hyperparameter $\alpha$. As noted in Section 2, we set $\alpha = 1$ in LoFT. In practice, the HuggingFace PEFT library implements scaling as $\alpha/r$, so setting $\alpha = r$ yields an effective scaling factor of 1, thereby removing the need for hyperparameter tuning. For LoRA and DoRA baselines, we followed the recommended setting $\alpha = 2r$ (see Table 10 in Liu et al. (2024)) to ensure fairness. One of LoFT's design goals is precisely to eliminate this hyperparameter, which we emphasize in the main text.

# D  LoFT ALGORITHM

---

**Algorithm 1** LoFT-AdamW with Alternating Updates

---

**Require:** Pretrained weights $W_0$, low-rank factors $U_0, V_0$, learning rate $\eta_k$, weight decay rate $\lambda$, AdamW parameters $\beta_1, \beta_2, \varepsilon$

1: Initialize first and second moments: $m_0^U, m_0^V, p_0^U, p_0^V \leftarrow 0$
2: Set alternating update flag: update_U $\leftarrow$ False
3: **for** $k = 1, 2, \ldots$ **do**
4:     $W_k \leftarrow W_0 + U_k V_k^\top$         # Reconstruct full weight matrix
5:     $g_W \leftarrow \nabla_W f(W_k)$         # Get full gradient (only for notational purposes)
6:     $g_U \leftarrow g_W V_k, \;\; g_V \leftarrow g_W^\top U_k$     # Project gradients to low-rank factors
7:     $C_k^V \leftarrow (V_{k-1}^\top V_k)(V_k^\top V_k)^{-1}, \quad C_k^U \leftarrow (U_{k-1}^\top U_k)(U_k^\top U_k)^{-1}$
8:     $\tilde{g}_U \leftarrow g_U (V_k^\top V_k)^{-1}, \tilde{g}_V \leftarrow g_V (U_k^\top U_k)^{-1}$
9:     $m_k^U \leftarrow \beta_1 m_{k-1}^U C_k^V + (1 - \beta_1)\tilde{g}_U$     # First moment calibration
10:    $m_k^V \leftarrow \beta_1 m_{k-1}^V C_k^U + (1 - \beta_1)\tilde{g}_V$
11:    $p_k^U \leftarrow \beta_2 p_{k-1}^U (C_k^V \otimes C_k^V) + (1 - \beta_2)(\tilde{g}_U \bullet \tilde{g}_U)$     # Second moment calibration
12:    $p_k^V \leftarrow \beta_2 p_{k-1}^V (C_k^U \otimes C_k^U) + (1 - \beta_2)(\tilde{g}_V \bullet \tilde{g}_V)$
13:    **if** update_U **then**         # Alternating updates
14:       $v_k^U \leftarrow p_k^U (V_k * V_k)$     # Reconstruct second moment in projected space
15:       $\tilde{m}_k^U \leftarrow m_k^U V_k^\top / (1 - \beta_1^k)$
16:       $\tilde{v}_k^U \leftarrow v_k^U / (1 - \beta_2^k)$
17:       $\Delta U \leftarrow \eta_k \cdot \dfrac{\tilde{m}_k^U}{\sqrt{\tilde{v}_k^U} + \varepsilon} V_k (V_k^\top V_k)^{-1}$     # Update $U$ with projection
18:       $U_{k+1} \leftarrow (1 - \lambda \eta_k) U_k - \Delta U$
19:       $V_{k+1} \leftarrow V_k$
20:    **else**
21:       $v_k^V \leftarrow p_k^V (U_k * U_k)$     # Reconstruct second moment in projected space
22:       $\tilde{m}_k^V \leftarrow m_k^V U_k^\top / (1 - \beta_1^k)$
23:       $\tilde{v}_k^V \leftarrow v_k^V / (1 - \beta_2^k)$
24:       $\Delta V \leftarrow \eta_k \cdot \dfrac{\tilde{m}_k^V}{\sqrt{\tilde{v}_k^V} + \varepsilon} U_k (U_k^\top U_k)^{-1}$     # Update $V$ with projection
25:       $V_{k+1} \leftarrow (1 - \lambda \eta_k) V_k - \Delta V$
26:       $U_{k+1} \leftarrow U_k$
27:    **end if**
28:    update_U $\leftarrow$ **not** update_U     # Alternate update direction
29: **end for**

---

## D.1  LoFT-MUON

In this section, we provide an extension of our approach to Muon (Jordan et al., 2024) algorithm. Firstly, we introduce the original Muon in Algorithm 2.

---

**Algorithm 2** Muon

---

**Require:** Learning rates $\eta_k$, momentum $\mu$

1: Initialize $m_0 \leftarrow 0$
2: **for** $k = 1, 2, \ldots$ **do**
3:     $g_W \leftarrow \nabla_W f(W_k)$     # Compute full gradient
4:     $m_k \leftarrow \mu m_{k-1} + g_W$     # Compute momentum
5:     $o_k \leftarrow \text{NewtonSchulz5}(m_k)$     # Algorithm 3
6:     $W_{k+1} \leftarrow W_k - \eta_k o_k$     # Update Parameters
7: **end for**

---

---

**Algorithm 3** NewtonSchulz5

---

**Require:** number of steps $n_{\text{steps}}$, $\varepsilon = 1e^{-7}$, $G \in \mathbb{R}^{m \times n}$, $(a, b, c) = (3.4445, -4.7750, 2.0315)$
  1: $X \leftarrow G/(\|G\|_F + \varepsilon)$                                                     # Proper initialization
  2: **if** m > n **then**                                                # For efficient computations
  3:     $X \leftarrow X^\top$
  4: **end if**
  5: **for** $k = 1, 2, \ldots, n_{\text{steps}}$ **do**
  6:     $A \leftarrow XX^\top$
  7:     $B \leftarrow bA + cA^2$
  8:     $X \leftarrow aX + BX$
  9: **end for**
10: **if** m > n **then**
11:     $X \leftarrow X^\top$
12: **end if**
13: **return** $X$

---

Examining the Muon algorithm, we observe that, like Adam, it employs first-order momentum; therefore, to adapt it to the LoFT setting, we can directly apply the first three building blocks. Furthermore, we can reconstruct an estimate of the full-finetuning momentum using (8), i.e., $\tilde{m}_k^U = m_k^U V_k^\top$. We note that the $\tilde{m}_k^U$ is at most rank $r$, but NewtonSchulz5 (Algortihm 3) can't take advantage of that and directly plugging in $\tilde{m}_k^U$ to NewtonSchulz5 would not benefit computations as we would be working with large $m \times n$ matrix. Therefore, we design efficient version of NewtonSchulz5 algortihm that accounts for low-rank inputs, see below.

---

**Algorithm 4** NewtonSchulz5_LowRank

---

**Require:** number of steps $n_{\text{steps}}$, $\varepsilon = 10^{-7}$, $U \in \mathbb{R}^{m \times r}$, $V \in \mathbb{R}^{n \times r}$ $(G = UV^\top)$, $(a, b, c) = (3.4445, -4.7750, 2.0315)$
  1: **if** $m > n$ **then**                               # For efficient computations (mirror of dense case)
  2:     $U, V \leftarrow V, U$                                      # Flip $U, V$
  3: **end if**
  4: $UtU \leftarrow U^\top U \in \mathbb{R}^{r \times r}$;    $VtV \leftarrow V^\top V \in \mathbb{R}^{r \times r}$
  5: $\|G\|_F \leftarrow \sqrt{\text{tr}((U^\top U)(V^\top V))}$                # Proper initialization (low-rank)
  6: $X_c \leftarrow \dfrac{1}{\|G\|_F + \varepsilon} I_r$              # Core $r \times r$ variable; $X = UX_cV^\top$
  7: **for** $k = 1, 2, \ldots, n_{\text{steps}}$ **do**
  8:     $S \leftarrow X_c VtV X_c^\top$                            # $r \times r$ form of $XX^\top$
  9:     $A \leftarrow SUtU$
10:     $B \leftarrow b\,A + c\,A^2$
11:     $X_c \leftarrow a\,X_c + BX_c$
12: **end for**
13: $X_U \leftarrow UX_c$          # $X \leftarrow UX_cV^\top$, we use only $UX_c$ as $V$ is added implicitly via $V_k^\top$ / $U_k(U/V\text{-update})$
14: **if** $m > n$ **then**
15:     $X_U \leftarrow VX_c^\top$
16: **end if**
17: **return** $X_U$         # Partial polar factors of $G$; cost per step $\mathcal{O}((m + n)r^2 + r^3)$

---

We note that the cost per step of this algorithm is only $\mathcal{O}((m + n)r^2 + r^3)$. Finally, we are ready to proceed with LoFT-Muon, which only requires extra memory of $\mathcal{O}((m + n)r)$, thus matching the standard LoRA memory requirements.

---

**Algorithm 5** LoFT-Muon

---

**Require:** learning rates $\eta_k$, momentum parameter $\mu$,
 1: Initialize $m_0^U, m_0^V \leftarrow 0$, weight decay rate $\lambda$
 2: Set alternating update flag: update_U $\leftarrow$ False
 3: **for** $k = 1, 2, \ldots$ **do**
 4:      # Reconstruct full weight matrix
 5:      $W_k \leftarrow W_0 + U_k V_k^\top$
 6:      # Get full gradient (only for notational purposes)
 7:      $g_W \leftarrow \nabla_W f(W_k)$
 8:      # Project gradients to low-rank factors
 9:      $g_U \leftarrow g_W V_k, \quad g_V \leftarrow g_W^\top U_k$
10:      $C_k^V \leftarrow (V_{k-1}^\top V_k)(V_k^\top V_k)^{-1}, \quad C_k^U \leftarrow (U_{k-1}^\top U_k)(U_k^\top U_k)^{-1}$
11:      $\tilde{g}_U \leftarrow g_U(V_k^\top V_k)^{-1}, \tilde{g}_V \leftarrow g_V(U_k^\top U_k)^{-1}$
12:      # First moment calibration
13:      $m_k^U \leftarrow \mu m_{k-1}^U C_k^V + \tilde{g}_U$
14:      $m_k^V \leftarrow \mu m_{k-1}^V C_k^U + \tilde{g}_V$
15:      # Alternating updates
16:      **if** update_U **then**
17:          $\Delta_U \leftarrow$ NewtonSchulz5_LowRank$(m_k^U, V_k)$
18:          $U_{k+1} \leftarrow (1 - \lambda\eta_k)U_k - \Delta U$
19:          $V_{k+1} \leftarrow V_k$
20:      **else**
21:          $\Delta_V \leftarrow$ NewtonSchulz5_LowRank$(m_k^V, U_k)$
22:          $V_{k+1} \leftarrow (1 - \lambda\eta_k)V_k - \Delta V$
23:          $U_{k+1} \leftarrow U_k$
24:      **end if**
25:      # Alternate update direction
26:      update_U $\leftarrow$ **not** update_U
27: **end for**

---

# E  OPTIMIZATION AND EFFICIENCY ANALYSIS

## E.1  ABLATION STUDY

In this ablation study, we investigate the contribution of key components in our proposed LoFT method by selectively disabling them and observing the impact on performance. The goal is to isolate the effectiveness of (i) state calibration, and (ii) alternate updates. The experiments are conducted on the WikiText-2 dataset using a GPT-2 model in a causal language modeling setup.

We evaluate four variants:

- **LoFT (full method)**: includes both alternate updates and state calibration.
- **LoFT without alternate updates**: removes the alternation mechanism while keeping calibration.
- **LoFT without state calibration**: disables calibration while retaining alternating updates.
- **LoFT without either**: disables both the alternation and state calibration.

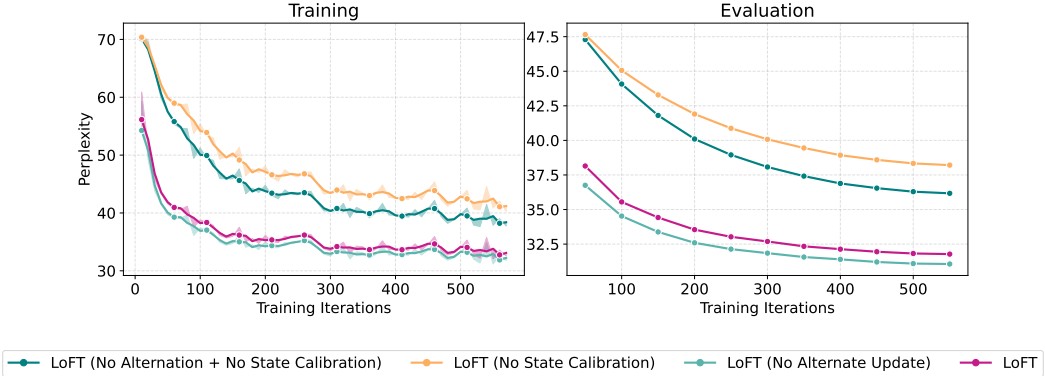

Figure 6: Ablation study of the proposed approach on a language modeling task. We train a GPT-2 model on the WikiText-2 dataset and evaluate the effect of key components of LoFT by incrementally removing (i) state calibration, (ii) alternate update, and (iii) both. Training perplexity (left) shows smoothed curves with shaded raw values, while evaluation perplexity (right) presents the unsmoothed results.

Training and evaluation perplexities are reported in Figure 6. For training curves, we show smoothed perplexity (3-step centered moving average) with raw values shaded underneath; evaluation perplexity is shown unsmoothed.

The best-performing variant in this specific setting is LoFT without alternate updates, which slightly outperforms the full LoFT setup. This is likely due to the fact that removing alternation effectively doubles the update frequency of LoFT parameters, which proves beneficial on WikiText-2 with GPT-2. We can see a significant decrease in performance when considering variants that do not have state calibration.

These results highlight the importance of state calibration, while they also suggest that LoFT can be slightly improved if we consider parallel updates. We attribute this to the small step size and gradient clipping, which limit the impact of the cross term that could be problematic in some cases.

**Scaling up to LLaMA and ViT.**   To further assess generality, we conduct additional ablations on one large language model benchmark and one vision benchmark. Specifically, we evaluate LoFT on LLaMA-7B with commonsense reasoning tasks and on ViT-Base for CIFAR-100 classification. In each case, we remove LoFT components one at a time. The results are reported in Tables 6 and 7.

Across both language and vision benchmarks, the results align with our GPT-2 findings. LoFT without alternation sometimes matches or slightly outperforms the full method, likely due to increased update frequency. In contrast, removing state calibration consistently causes large performance drops, particularly dramatic on LLaMA-7B. Overall, these ablations confirm that both alternation and state

Table 6: Ablation results on CIFAR-100 with ViT-Base.

| LoFT variant | Accuracy |
|---|---|
| Full method | 91.18 |
| No alternation | **91.60** |
| No state calibration | 89.38 |
| No alternation + no state calibration | 89.94 |

Table 7: Ablation results on LLaMA-7B with commonsense reasoning benchmarks.

| LoFT Variant | BoolQ | PIQA | SIQA | HS | WG | ARC-C | ARC-E | OBQA | avg. |
|---|---|---|---|---|---|---|---|---|---|
| Full method | 67.34 | 80.96 | 76.20 | 80.50 | 76.40 | 63.62 | 79.21 | 75.40 | **74.95** |
| No alternation | 68.56 | 79.26 | 77.33 | 77.16 | 78.37 | 62.97 | 79.42 | 74.40 | 74.68 |
| No state calibration | 03.18 | 48.80 | 66.43 | 21.17 | 68.90 | 55.03 | 75.63 | 66.00 | 50.64 |
| No alt. + no state cal. | 57.31 | 62.24 | 55.58 | 17.27 | 65.27 | 56.48 | 73.06 | 68.60 | 56.98 |

calibration are important contributors, with calibration being indispensable for LoFT's stability and effectiveness.

### E.2 MEMORY FOOTPRINT

We evaluate the memory efficiency of LoFT in comparison to LoRA, DoRA, and DoRA (simple) under two configurations: rank $r=16$ and rank $r=4$. All experiments were conducted using the LLaMA-7B model on commonsense reasoning tasks (Tables 8 and 9).

**Theoretical analysis.** For AdamW, LoRA requires

$$mn\,[W_0] + (m+n)r\,[U,V] + 4(m+n)r\,[\text{optimizer states}] \;=\; mn + 5(m+n)r,$$

while LoFT requires

$$
\begin{aligned}
mn\,[W_0] \;+\;& (m+n)r\,[U,V] + 2(m+n)r\,[\text{previous iterates}] + 2(m+n)r\,[\text{momentum}] \\
+\;& 2(m+n)r^2\,[\text{cross-terms}] = mn + 5(m+n)r + 2(m+n)r^2.
\end{aligned}
$$

The additional $2(m+n)r^2$ term arises from cross-terms for optimizer state recalibration. Crucially, this scales with $(m+n)$ rather than $mn$, ensuring LoFT remains far more efficient than full fine-tuning when $r$ is small.

**Empirical results.** At rank $r=16$, LoFT matches LoRA in terms of trainable parameter percentage (0.4145%) with no increase, while incurring only a $\underline{+25.65\%}$ increase in memory usage. This memory cost is nearly identical to DoRA (simple), which also maintains a low overhead (+**25.23**%), and significantly lower than full DoRA, which increases memory by over 341%.

At the lower rank setting $r=4$, LoFT maintains parameter parity with LoRA (0.1040%) and achieves a very modest memory increase of just +**6.71**%, compared to the large 342% increase with DoRA. While DoRA (simple) also limits memory to some extent, it still shows over 25% overhead and increases trainable parameters by 12.4%.

Table 8: Comparison of trainable parameter percentage and memory usage for different methods at rank $r=16$ using LLaMA-7B on commonsense reasoning tasks.

| Method | Trainable params (%) | + Relative Incr. | Memory (GB) | + Relative Incr. |
|---|---|---|---|---|
| LoRA | 0.4145 | +0.00% | 28.50 | +0.00% |
| DoRA | 0.4274 | +3.11% | 125.95 | +341.93% |
| DoRA (simple) | 0.4274 | +3.11% | 35.69 | +**25.23**% |
| LoFT | 0.4145 | +**0.00**% | 35.81 | $\underline{+25.65\%}$ |

Table 9: Comparison of trainable parameter percentage and memory usage for different methods at rank $r$=4 using LLaMA-7B on commonsense reasoning tasks.

| Method | Trainable params ($\%$) | + Relative Incr. | Memory (GB) | + Relative Incr. |
|---|---|---|---|---|
| LoRA | 0.1040 | +0.00% | 28.15 | +0.00% |
| DoRA | 0.1169 | +12.40% | 124.47 | +342.17% |
| DoRA (simple) | 0.1169 | +12.40% | 35.27 | +25.29% |
| LoFT | 0.1040 | **+0.00%** | 30.04 | **+6.71%** |

**Accuracy-efficiency trade-off.** Although LoFT introduces extra memory overhead relative to LoRA, it achieves higher accuracy at substantially smaller ranks. For instance, LoFT with $r = 4$ surpasses LoRA with $r = 16$ on LLaMA-7B, and LoFT with $r = 1$ surpasses LoRA with $r = 16$ on LLaMA-2-7B (see Table 2). Importantly, unlike DoRA, LoFT adds no backward-pass memory overhead (cf. Section 4.3 in DoRA (Liu et al., 2024)). Thus, the modest increase is offset by substantial performance gains at lower ranks.

**LoFT (simple).** We further identify that the main bottleneck in LoFT stems from the second-moment calibration. Since its effect on accuracy is marginal ($\sim 0.1\%$ drop at LLaMA-7B, $r$=16) (Table 10), we propose *LoFT (simple)*, which omits this step. As shown in Table 11, LoFT (simple) reduces overhead to under $6\%$ compared to LoRA, while maintaining nearly identical accuracy.

Table 10: Performance comparison between LoFT and LoFT (simple) (LLaMA-7B, rank $r = 16$).

| Method | BoolQ | PIQA | SIQA | HS | WG | ARC-C | ARC-E | OBQA | avg. |
|---|---|---|---|---|---|---|---|---|---|
| LoFT | 68.62 | 82.80 | 78.27 | 82.69 | 73.32 | 64.30 | 80.26 | 78.40 | 76.08 |
| LoFT (simple) | 68.50 | 81.18 | 78.20 | 76.87 | 78.93 | 64.85 | 81.14 | 78.20 | 75.98 |

Table 11: LoFT (simple) memory overhead on LLaMA-7B under ranks $r = 16$ and $r = 4$.

| Method | Memory (GB) | + Relative Incr. |
|---|---|---|
| LoRA | 28.50 | +0.00% |
| LoFT (simple) ($r = 16$) | 30.02 | +5.35% |
| LoFT (simple) ($r = 4$) | 29.61 | +5.18% |

Overall, LoFT offers the same parameter efficiency as LoRA while delivering competitive performance with substantially lower memory demands than DoRA variants. This makes LoFT a memory-efficient alternative suitable for deployment in resource-constrained settings.

We refer the reader to (Liu et al., 2024) for detailed definitions of DoRA and DoRA (simple). In our experiments, we exclusively used DoRA (simple), as recommended by DoRA's authors. Also, the full DoRA implementation requires substantially more memory and is impractical to run on one GPU.

E.3 TRAINING LATENCY

We also evaluate the training latency of LoFT relative to LoRA and DoRA (simple). LoFT introduces additional overhead due to optimizer state alignment and recalibration. Table 12 summarizes the latency (forward + backward + optimizer step) on LLaMA-7B across different ranks.

To better understand the main sources of overhead, we conducted ablations of LoFT variants at rank $r$=4. The results are shown in Table 13.

These results indicate that the main bottleneck of LoFT arises from second-moment calibration. Omitting this step yields LoFT (simple), which reduces latency to within $\sim 30\%$ of LoRA while being $\sim 2\times$ faster than the stronger baseline DoRA.

Table 12: Relative training latency of LoRA, DoRA (simple), LoFT, and LoFT (simple) on LLaMA-7B. Latency is reported as a multiplicative factor relative to LoRA.

| Method | $r = 16$ | $r = 4$ | $r = 1$ |
|---|---|---|---|
| LoRA | $1.00\times$ | $1.00\times$ | $1.00\times$ |
| DoRA (simple) | $2.38\times$ | $2.54\times$ | $2.54\times$ |
| LoFT | $3.23\times$ | $2.26\times$ | $1.76\times$ |
| LoFT (simple) | $\mathbf{1.32\times}$ | $\mathbf{1.27\times}$ | $\mathbf{1.22\times}$ |

Table 13: Latency breakdown (in seconds) for LoFT variants at rank $(r = 4)$ (LLaMA-7B).

| LoFT Variant | Latency (s) |
|---|---|
| LoFT (full) | 1.0903 |
| No alternation | 1.5703 |
| No state calibration | 0.5810 |
| No alternation + no state calibration | 0.6146 |
| No second moment calibration [LoFT (simple)] | 0.6265 |

**Implementation note.** All latency measurements are based on a plain PyTorch implementation. We expect substantial speed-ups with a dedicated CUDA kernel implementations, which we plan as future work.

### E.4 COMPARISON WITH ADDITIONAL BASELINES

**Experimental Setup.** We first fine-tune the original **GPT-2** (137M) on WikiText2 using the same data split and preprocessing as Radford et al. (2019). All methods share the same training hyperparameters: 1 epoch, AdamW optimizer, batch size 64, learning rate $2\times10^{-4}$ with linear decay. For adapter-based baselines (Hu et al., 2022; Kalajdzievski, 2023; Zhang et al., 2023; Wang et al., 2024; 2025) we set the rank $r=4$; LoRA$^+$ uses its default temperature and dropout as in the official repository. After convergence, we evaluate on the WikiText-2 validation set and report *perplexity* (lower is better).

**Limitations of certain baselines.** VeRA (Kopiczko et al., 2024) and DoRA (Liu et al., 2024) only handle `Linear` layers. Because GPT-2 implements attention weights as `Conv1D` layers, reproducing these methods would require serious surgery and a major rewrite; we, therefore, omit them. In practice, this means VeRA and DoRA cannot be applied unchanged to a large family of models that rely on `Conv1D` parameterizations.

Table 14: Perplexity (PPL) results on the WikiText2 dataset for various fine-tuning methods applied to GPT-2. Lower values indicate better performance. LoFT achieves the best result, outperforming other parameter-efficient techniques.

| Model | Method | WikiText2 (PPL ↓) |
|---|---|---|
| | Zero-Shot | 60.38 |
| | Full FT | **29.51** |
| **GPT-2** | LoRA$_{r=4}$ | 34.80 |
| | rsLoRA$_{r=4}$ | 32.96 |
| | AdaLoRA$_{r=4}$ | 55.67 |
| | LoRA-Pro$_{r=4}$ | 32.79 |
| | LoRA-GA$_{r=4}$ | 37.34 |
| | LoRA+$_{r=4}$ | 36.15 |
| | LoFT$_{r=4}$ | 31.75 |

Table 15: Perplexity (PPL) on WikiText2 for GPT-2 Large using various fine-tuning methods. LoFT achieves the best performance, outperforming full fine-tuning and other parameter-efficient techniques.

| Model | Method | WikiText2 (PPL ↓) |
|---|---|---|
| | Zero-Shot | 38.87 |
| | Full FT | 19.42 |
| **GPT-2 Large** | LoRA$_{r=4}$ | 19.78 |
| | rsLoRA$_{r=4}$ | 19.62 |
| | AdaLoRA$_{r=4}$ | 23.31 |
| | LoRA-Pro$_{r=4}$ | 20.06 |
| | LoRA-GA$_{r=4}$ | 21.44 |
| | LoRA+$_{r=4}$ | 19.73 |
| | LoFT$_{r=4}$ | **19.26** |

**Results on GPT-2.** Table 14 reports validation perplexity. LoFT yields the lowest PPL (31.75), outperforming all other parameter-efficient baselines and coming within 2.2 points of full fine-tuning while updating only a small fraction of parameters. AdaLoRA (Zhang et al., 2023) performs noticeably worse in this low-resource regime. Training and evaluation curves are visualized in Figure 7: LoFT converges smoothly and tracks Full FT closely throughout training, whereas other methods plateau higher.

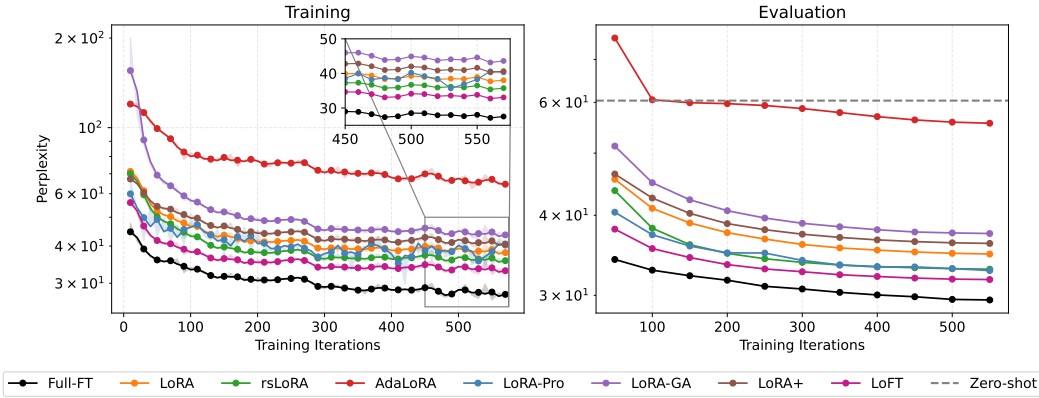

Figure 7: Training and evaluation perplexity curves for GPT-2 on WikiText-2 dataset. The left panel shows smoothed training perplexity (3-point moving average) for seven fine-tuning methods (Full-FT, LoRA, rsLoRA, AdaLoRA, LoRA-Pro, LoRA-GA, LoRA$^+$, and LoFT), with the raw PPL shaded beneath each curve. The right panel reports evaluation PPL for the same methods, with a dashed horizontal line at 60.38 marking the zero-shot baseline. Table for a reference: Table 14.

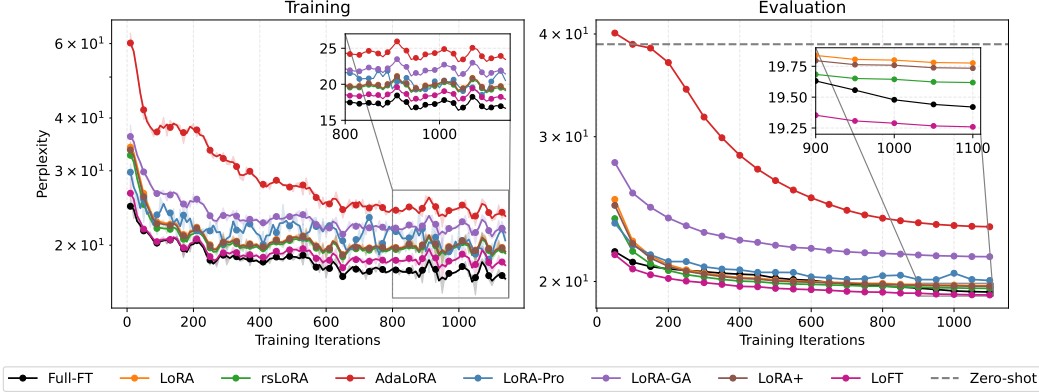

Figure 8: Training and evaluation perplexity curves for GPT-2 LARGE on WikiText-2 dataset. The left panel shows smoothed training perplexity (3-point centered moving average) for seven fine-tuning methods (Full-FT, LoRA, rsLoRA, AdaLoRA, LoRA-Pro, LoRA-GA, LoRA$^+$, and LoFT). The right panel presents evaluation perplexity curves, with a dashed horizontal line at 38.87 marking the zero-shot baseline. Table for a reference: Table 15.

**Scaling to GPT-2 Large.** We repeat the experiment on **GPT-2 Large** (812M) with the same data and hyper-parameters (batch size reduced to 32 to fit memory for full fine-tuning). Table 15 extends the comparison to this larger model. The zero-shot model perplexity is 38.87. Full fine-tuning brings this down to 19.42, but LoFT achieves an even lower 19.26 while updating only a small fraction of weights. The other adapter-style baselines cluster a few tenths higher (LoRA 19.78, rsLoRA 19.62, LoRA$^+$ 19.73), and AdaLoRA again lags behind at 23.31. In relative terms, LoFT improves on the vanilla LoRA baseline by 2.6% and narrows (indeed, slightly surpasses) the gap to full fine-tuning, confirming that the gains observed on the smaller GPT-2 model persist and even strengthen at a larger scale.

Figure 8 highlights an interesting trend: on GPT-2 Large, Full FT achieves the lowest *training* perplexity, but its *evaluation* perplexity stalls above LoFT, evidence of *overfitting* as model capacity grows. By contrast, the low-rank structure of LoFT provides a built-in regularizer: it follows Full-FT during training yet generalizes better, maintaining leading evaluation PPL. On the smaller GPT-2 (137M), Full-FT still wins on both train and evaluation – there, capacity is not large enough to overfit the WikiText2 dataset – whereas at 812M parameters, the risk of memorization rises and LoFT's parameter-efficient updates prove more robust.

# F    LANGUAGE MODEL EXPERIMENTS

## F.1    COMMONSENSE REASONING

For completeness, Table 16 provides the exact task-wise accuracy scores for all methods and rank settings shown in Figure 4 of the main paper. These results quantify how LoRA, DoRA, and LoFT behave across eight commonsense reasoning benchmarks when applied to LLaMA-7B with rank $r \in \{4, 2, 1\}$.

As noted in the main text, LoFT maintains high and stable accuracy across all tasks, even under extreme compression (rank 1), whereas both LoRA and DoRA degrade substantially – especially on more complex tasks like HellaSwag (HS), Winogrande (WG), and SIQA. Notably, DoRA at $r=4$ and $r=2$ exhibits drastic task-level failures, with near-zero performance on WG and erratic behavior across others, reflecting instability under constrained adaptation. In contrast, LoFT consistently performs well across ranks, confirming its robustness under limited parameter budgets.

See Table 16 for the exact per-task numbers.

Table 16: Task-wise performance of LoRA, DoRA, and LoFT on commonsense reasoning benchmarks at lower ranks ($r = \{4, 2, 1\}$) using LLaMA-7B. While LoFT maintains stable accuracy across all tasks, both LoRA and DoRA show significant drops – particularly on complex benchmarks such as HellaSwag and Winogrande – indicating their limited reliability under extreme parameter constraints.

| Model | Method | BoolQ | PIQA | SIQA | HS | WG | ARC-C | ARC-E | OBQA | avg. |
|---|---|---|---|---|---|---|---|---|---|---|
| | LoRA$_{r=4}$ | 66.15 | 43.47 | 42.12 | 24.46 | 72.85 | 47.18 | 53.03 | 48.80 | 49.76 |
| | LoRA$_{r=2}$ | 67.77 | 66.50 | 40.63 | 21.85 | 53.28 | 50.26 | 63.51 | 52.00 | 51.97 |
| | LoRA$_{r=1}$ | 66.15 | 74.05 | 73.58 | 35.24 | **77.19** | 59.56 | 76.43 | 70.80 | 66.62 |
| | DoRA$_{r=4}$ | 32.35 | 7.13 | 47.03 | 27.54 | 0.00 | 52.65 | 66.37 | 46.60 | 34.96 |
| LLaMA-7B | DoRA$_{r=2}$ | 57.55 | 70.38 | **76.41** | 48.55 | 9.71 | 62.03 | 78.66 | **75.40** | 59.84 |
| | DoRA$_{r=1}$ | 67.16 | 77.26 | 76.25 | 31.38 | 20.60 | 57.34 | 70.50 | 64.00 | 58.06 |
| | LoFT$_{r=4}$ | 67.34 | **80.96** | 76.20 | **80.50** | 76.40 | 63.62 | **79.21** | 75.40 | **74.95** |
| | LoFT$_{r=2}$ | **68.03** | 79.16 | 75.84 | 78.86 | 76.24 | **64.51** | 78.03 | 71.00 | 73.96 |
| | LoFT$_{r=1}$ | 67.09 | 78.35 | 74.46 | 76.14 | 74.82 | 58.87 | 76.85 | 70.80 | 72.17 |

## F.2    MATHEMATICAL REASONING AND QUANTIZED LoFT

**Setup.**    We evaluate exact-match accuracy on the Orca-Math dataset (Mitra et al., 2024) using LLaMA2 and LLaMA3 models. Our experimental setup is largely based on the QLoRA fine-tuning recipe outlined by Answer.ai (Turgutlu et al., 2024), with a few key modifications. Specifically, we quantize the pre-trained model to 4-bit and fine-tune each model for 3 epochs on 200k training examples using bf16 precision, a global batch size of 32, the AdamW optimizer, and a shortened context window of 256 tokens. Evaluation is performed on 500 held-out examples using exact-match comparison, following the original methodology. We adopt the zero-shot and five-shot prompting results directly from the blog post: for LLaMA2, these are 0.07 and 0.08, and for LLaMA3, 0.23 and 0.27, respectively.

For parameter-efficient fine-tuning, we compare QLoRA (Dettmers et al., 2023) with our proposed method, QLoFT – a quantized variant of LoFT designed for greater efficiency. We evaluate QLoRA at a fixed rank of 16, yielding 0.15 accuracy on LLaMA2 and 0.292 accuracy on LLaMA3. Under the same rank ($r=16$), QLoFT achieves higher accuracy: 0.16 on LLaMA2 and 0.324 on LLaMA3. To assess robustness under constrained parameter budgets, we further reduce QLoFT's rank to 8, 4

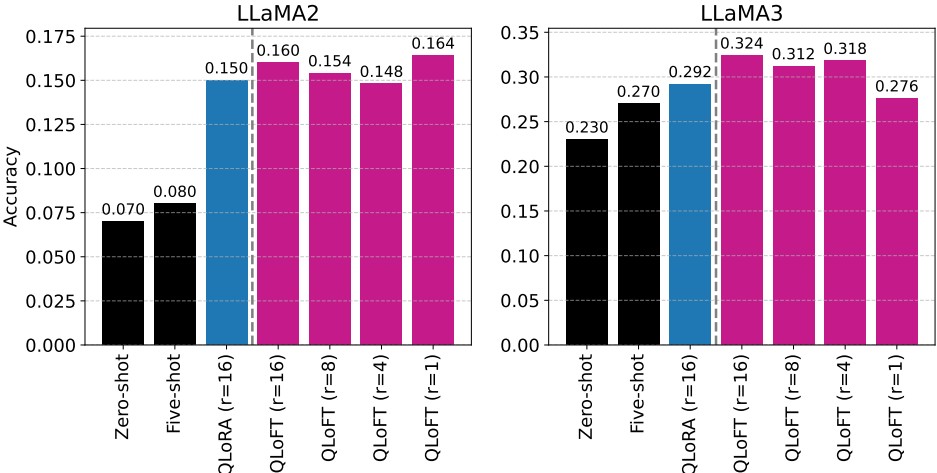

Figure 9: Accuracy comparison on the Orca-Math dataset using LLaMA2 and LLaMA3 models. We compare our method, QLoFT, quantized version of LoFT, with QLoRA. QLoFT is evaluated at various ranks ($r = \{16, 8, 4, 1\}$) and consistently outperforms QLoRA, demonstrating superior performance in parameter-efficient fine-tuning for mathematical reasoning.

and 1. Even with 75% fewer trainable parameters ($r=4$), QLoFT maintains strong performance – 0.148 on LLaMA2 and 0.318 on LLaMA3 – matching or exceeding QLoRA's results. At $r=1$, it still performs competitively, reaching 0.164 on LLaMA2 and 0.276 on LLaMA3.

Overall, QLoFT consistently outperforms QLoRA at equivalent ranks across both model backbones, demonstrating better adaptation capacity with identical parameter budgets. More importantly, the performance drop as the rank decreases is surprisingly small, highlighting QLoFT's ability to retain strong accuracy even in highly constrained regimes. On LLaMA3, the benefits are even more pronounced: QLoFT outperforms QLoRA by over 3 points at $r=16$, and continues to lead at $r=\{8, 4\}$. This suggests that QLoFT better leverages the capacity of larger models, effectively leveraging their increased capacity for improved tuning.

### F.3 SCALING TO LLAMA-3-70B

To further evaluate the scalability of LoFT, we conduct experiments on the substantially larger LLaMA-3-70B model. These experiments follow the same unified training protocol used in the main commonsense reasoning evaluation. All hyperparameters follow the same configuration as explained in Section C.2, with varying rank $r$ and the learning rate set to $5 \times 10^{-3}$. Table 17 reports the results for LoFT at ranks $r \in \{16, 4, 2, 1\}$. These results confirm that the robustness trends observed on smaller backbones persist at the 70B scale, even at rank 1.

Table 17: Commonsense reasoning results on LLaMA-3-70B using LoFT at varying ranks.

| Method | BoolQ | PIQA | SIQA | HS | WG | ARC-C | ARC-E | OBQA | avg. |
|---|---|---|---|---|---|---|---|---|---|
| LoFT ($r = 16$) | 79.45 | 92.82 | 82.24 | 97.02 | 90.06 | 92.75 | 97.52 | 93.40 | 90.66 |
| LoFT ($r = 4$) | 79.17 | 92.60 | 81.58 | 96.70 | 89.34 | 92.06 | 97.31 | 92.60 | 90.17 |
| LoFT ($r = 2$) | 78.38 | 92.55 | 81.78 | 96.61 | 88.40 | 92.06 | 97.31 | 92.40 | 89.94 |
| LoFT ($r = 1$) | 77.83 | 91.51 | 80.35 | 95.79 | 86.03 | 91.47 | 97.05 | 91.40 | 88.93 |

## G VISION EXPERIMENTS

### G.1 TRAINING DYNAMICS

In Figure 5 of the main paper, we presented the training performance curves on the HAM10000 dataset. Here, in Appendix Figure 10, we show analogous training-loss dynamics (log scale) for the

three remaining image-classification benchmarks: ISIC2019, Diabetic Retinopathy, and DomainNet. Each panel plots the raw per-step loss ($\alpha=0.25$) beneath a 10-step centered moving average, with a zoomed inset in the upper-right corner of the latter two datasets to highlight differences in the final epochs.

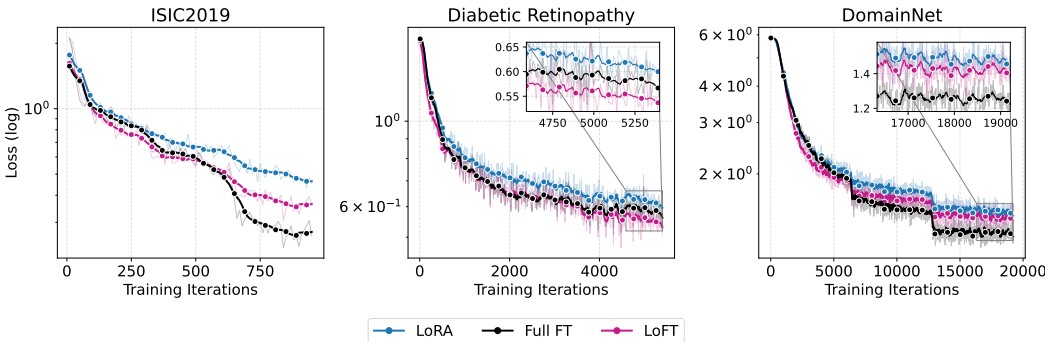

Figure 10: Additional training-loss dynamics for image classification. For the remaining benchmarks, ISIC2019 (left), Diabetic Retinopathy (center), and DomainNet (right), we plot training loss. **LoFT (magenta)** consistently outperforms **LoRA (blue)** and closely tracks **full fine-tuning (black)**, achieving the lowest loss on Diabetic Retinopathy and substantially narrowing the gap on ISIC2019 and DomainNet. See Figure 5 in the main paper for the HAM10000 curves.

Across all three tasks, LoFT (magenta) consistently outperforms LoRA (blue) and closes much of the gap to full fine-tuning (black). In particular:

- **Diabetic Retinopathy**: LoFT achieves the lowest training loss of all three methods throughout, demonstrating its strongest advantage in this medical imaging dataset.
- **ISIC2019 & DomainNet**: LoFT again reduces loss more quickly than LoRA and tracks very closely to full fine-tuning, especially in the later stages. While full FT still attains the absolute minimum loss, LoFT narrows the difference relative to LoRA.

## G.2 DOMAINNET: DOMAIN-SPECIFIC RESULTS

We would like to include the extended results of the experiment on the DomainNet dataset, including domain-specific performance results.

Table 18 complements the cross-dataset comparison in Table 3 (main paper) by breaking down the DomainNet dataset results by domain (*clipart*, *infograph*, *painting*, *quickdraw*, *real*, and *sketch*). All runs use the same ViT-Base backbone and optimization protocol described in Section C.

Table 18: Domain-specific accuracy results on the DomainNet dataset. While overall DomainNet results are presented in the main paper, this table provides detailed per-domain accuracy for various parameter-efficient fine-tuning methods.

| Model | Method | DomainNet Dataset | | | | | | |
|-------|--------|---------|-----------|----------|-----------|-------|--------|-------|
| | | clipart | infograph | painting | quickdraw | real | sketch | avg |
| | Full FT | **78.92** | **44.09** | **73.11** | **69.15** | 83.92 | **69.00** | **69.70** |
| | LoRA$_{r=16}$ | 77.64 | 42.86 | 72.44 | 66.59 | 84.50 | 67.21 | 68.54 |
| ViT-Base | DoRA$_{r=16}$ | 73.15 | 40.14 | 69.46 | 60.83 | 82.60 | 64.38 | 65.09 |
| | LoFT$_{r=16}$ | 78.11 | 42.95 | 72.80 | 68.10 | **84.55** | 68.37 | 69.15 |
| | LoFT$_{r=8}$ | 76.77 | 42.04 | 71.56 | 65.99 | 84.30 | 67.09 | 67.96 |
| | LoFT$_{r=4}$ | 73.38 | 40.15 | 69.58 | 60.98 | 82.83 | 64.10 | 65.17 |

The overall DomainNet numbers reported in Table 3 already show that LoFT$_{r=16}$ narrows the gap to Full FT and outperforms both LoRA and DoRA. However, DomainNet's six domains differ markedly

in style and label distribution; the per-domain breakdown reveals how each method copes with this heterogeneity.

**Main observations:**

- Full fine-tuning remains strongest on average $(69.7\%)$, topping five of six domains.
- LoFT with $r{=}16$ trails Full FT by only **0.55pp** on average and surpasses the full FT on the real domain $(84.55\%)$.
- LoRA lags LoFT on every domain except real, where both methods are statistically tied.
- DoRA and low-rank LoFT variants $(r{=}\{8, 4\})$ show the expected accuracy drop, but LoFT retains at least parity with the corresponding LoRA/DoRA settings.

In the main paper, we reported validation-set accuracy to keep the test labels unseen. For the extended analysis here we evaluate on the official test split (176743 images) to give a complete picture of domain-level generalization. No hyper-parameters were tuned on the test set; models are exactly those used in the main paper.

## G.3   COMPARISON WITH VISION-CENTRIC BASELINES

We further compare LoFT against two representative vision adaptation baselines: Visual Prompt Tuning (VPT) (Jia et al., 2022) and ViT-Adapter (Houlsby et al., 2019; Chen et al., 2023). To cover both standard and challenging scenarios, we evaluate on CIFAR-100 (general image classification) (Krizhevsky et al., 2009) and Diabetic Retinopathy (a medical imaging dataset) (Graham, 2015). Table 19 summarizes the results.

Table 19: Comparison of LoFT with VPT and ViT-Adapter on CIFAR-100 and Diabetic Retinopathy datasets using ViT-Base.

| Method | CIFAR-100 | Diabetic Retinopathy |
|---|---|---|
| Adapter | 85.93 | 51.82 |
| VPT | 90.03 | 50.28 |
| LoFT | **91.20** | **58.49** |

LoFT achieves the best performance across both datasets. While VPT and Adapter perform well on CIFAR-100, they struggle on the more complex medical dataset, whereas LoFT maintains strong results in both settings.

## H   CODING EXPERIMENTS

To further evaluate LoFT on representative large language model benchmarks for code generation, we conduct additional experiments on HumanEval and HumanEval+ (Chen et al., 2021). We use LLaMA3-8B as the backbone model and fine-tune it on the CodeFeedback (Python-only) dataset (Zheng et al., 2024), which consists of curated instruction-response pairs focused on Python code completion and correction. All experiments follow the same training recipe described in the main paper; the only variable changed across configurations is the rank $r$ of the low-rank adaptation. Evaluation is performed using the EvalPlus framework (Liu et al., 2023). For each problem, we sample 10 completions at temperature $T = 0.5$ and compute pass@k for $k \in \{1, 5, 10\}$. HumanEval+ is a stricter version of HumanEval with significantly expanded unit tests per task, providing a more rigorous assessment of functional correctness.

We first study the effect of increasing the rank beyond the range $r \leq 32$ considered in the main paper. Table 20 reports results for LoFT (simple) at $r \in \{8, 16, 32, 64, 128, 256\}$ and compares against LoRA under identical training and evaluation settings. At low rank ($r = 8$), LoFT already achieves strong performance, with pass@1 of 0.460 on HumanEval and 0.405 on HumanEval+. As the rank increases, performance improves modestly, reaching peak pass@1 at $r = 128$ (0.485 on HumanEval and 0.436 on HumanEval+), before flattening at $r = 256$. For broader sampling metrics,

Table 20: Rank scaling results on HumanEval and HumanEval+.

| Method | rank $r$ | HumanEval | | | HumanEval+ | | |
|---|---|---|---|---|---|---|---|
| | | pass@1 | pass@5 | pass@10 | pass@1 | pass@5 | pass@10 |
| LoFT (simple) | 8 | 0.460 | 0.682 | 0.738 | 0.405 | 0.624 | 0.683 |
| | 16 | 0.455 | 0.674 | 0.744 | 0.409 | 0.623 | 0.695 |
| | 32 | 0.468 | 0.673 | 0.732 | 0.423 | 0.619 | 0.679 |
| | 64 | 0.465 | 0.684 | 0.768 | 0.415 | 0.626 | 0.707 |
| | 128 | 0.485 | 0.678 | 0.750 | 0.436 | 0.627 | 0.697 |
| | 256 | 0.482 | 0.665 | 0.736 | 0.432 | 0.623 | 0.681 |
| LoRA | 8 | 0.421 | 0.631 | 0.720 | 0.368 | 0.572 | 0.659 |
| | 64 | 0.463 | 0.670 | 0.738 | 0.416 | 0.622 | 0.701 |

Table 21: Context length comparison at fixed rank $r = 8$. LoFT consistently outperforms LoRA across both context settings and both benchmarks.

| Method | Context | HumanEval | | | HumanEval+ | | |
|---|---|---|---|---|---|---|---|
| | | pass@1 | pass@5 | pass@10 | pass@1 | pass@5 | pass@10 |
| LoRA | 256 | 0.421 | 0.631 | 0.720 | 0.368 | 0.572 | 0.659 |
| LoFT | 256 | 0.482 | 0.681 | 0.756 | 0.439 | 0.614 | 0.689 |
| LoRA | 512 | 0.433 | 0.640 | 0.695 | 0.380 | 0.583 | 0.646 |
| LoFT | 512 | 0.488 | 0.674 | 0.738 | 0.445 | 0.612 | 0.671 |

pass@10 reaches its highest value at $r = 64$ on both benchmarks (0.768 on HumanEval and 0.707 on HumanEval+). These results indicate clear diminishing returns as rank increases, showing that small ranks already capture most of the achievable gains.

Importantly, LoFT demonstrates strong efficiency in the low-rank regime. At matched rank $r = 8$, LoFT outperforms LoRA (0.460 vs. 0.421 pass@1 on HumanEval, and 0.405 vs. 0.368 on HumanEval+). Notably, LoFT at $r = 8$ matches or exceeds the performance of LoRA at $r = 64$, indicating that LoFT achieves comparable accuracy with a substantially smaller rank. This highlights improved expressivity per parameter and better optimization alignment in the low-rank setting.

We further examine the effect of context length by evaluating both LoFT and LoRA at sequence lengths of 256 and 512 tokens with a fixed rank $r = 8$. Results are shown in Table 21. LoFT consistently outperforms LoRA across both context lengths and both benchmarks. On HumanEval, LoFT achieves pass@1 of 0.482 and 0.488 at context lengths 256 and 512, respectively, compared to 0.421 and 0.433 for LoRA. On HumanEval+, we observe a similar trend. The improvements remain stable across context windows, suggesting that the gains are not sensitive to moderate changes in sequence length.

