# OpenReview forum: "LoFT: Low-Rank Adaptation That Behaves Like Full Fine-Tuning"
_ICLR.cc/2026/Conference — ICLR 2026 Poster_

### Official Review · Reviewer_vRxF · 2025-10-26

**Soundness:** 3
**Presentation:** 3
**Contribution:** 3
**Rating:** 6
**Confidence:** 3

**Summary:**

This paper introduces LoFT, a new parameter-efficient fine-tuning method designed to make low-rank adaptation behave like full fine-tuning. Existing LoRA-based approaches mainly focus on gradient approximation but ignore the optimizer state misalignment, particularly in the first and second moments of the AdamW optimizer. LoFT explicitly aligns both gradients and optimizer states with full fine-tuning dynamics through six carefully designed components: alternating updates, gradient scaling, optimizer state calibration, second-moment alignment, projected full update reconstruction, and gradient clipping. Theoretical analysis proves that LoFT reduces exactly to AdamW when the rank equals the full dimension. Experiments on multiple large language models and vision transformers demonstrate that LoFT achieves higher accuracy and faster convergence than LoRA and DoRA while maintaining the same inference cost and number of trainable parameters.

**Strengths:**

1.	The method directly addresses the optimizer state misalignment problem that has been largely overlooked in prior low-rank adaptation research.
2.	The theoretical analysis is rigorous and provides a clear guarantee that LoFT degenerates to AdamW in the full-rank case.
3.	The six-component design is systematic, and is validated through detailed ablation studies.
4.	LoFT consistently outperforms LoRA and DoRA on both natural language and vision benchmarks, showing broad applicability.

**Weaknesses:**

1.	The additional memory cost, which can reach about twenty-five percent compared to LoRA, is not fully analyzed for its impact on large-scale training.
2.	Experiments are limited to models of eight billion parameters or smaller, leaving scalability to larger models unverified.
3.	The effect of optimizer state projection on stability and convergence speed is discussed conceptually but lacks quantitative analysis.
4.	The paper does not report concrete throughput or training speed measurements compared with LoRA or DoRA.

**Questions:**

1. What is the quantitative impact of the additional memory requirement on training efficiency and GPU utilization?

2. Can LoFT be extended to other optimizers such as Muon that use different moment estimation mechanisms?

---

> ### Author Response · Authors · 2025-11-21
>
> We sincerely thank the reviewer for their positive and constructive feedback. We appreciate the recognition of our contributions and are glad that the significance of our work was well received. Let us further clarify the mentioned weaknesses and questions below.
>
> > The additional memory cost, which can reach about twenty-five percent compared to LoRA, is not fully analyzed for its impact on large-scale training.
>
>  Thank you for this important comment. We agree that LoFT introduces additional memory overhead due to the storage of previous iterates and cross-terms required for recalibrating the optimizer state. This is an unavoidable cost when approximating projected full fine-tuning (see Figure 2).
> However, LoFT achieves higher accuracy at considerably smaller ranks, effectively offsetting the memory overhead. For instance, LLaMA-7B with LoFT (r=4) surpasses LoRA (r=16), and LLaMA-2-7B with LoFT (r=4) exceeds LoRA (r=16) (see Table 2). Unlike DoRA (the current SOTA), LoFT adds no backward-pass memory overhead (cf. Section 4.3 in DoRA). As shown in Appendix E.5, LoFT increases memory by up to ~25% compared to LoRA, but remains comparable to, or lower than, DoRA, while delivering consistently higher accuracy across ranks.
> Finally, to further reduce memory usage, we include LoFT (simple), a variant that omits second-moment calibration, the main source of additional memory, while incurring only ~0.1 % accuracy degradation. This reduces the overhead to < 6 % compared to LoRA while maintaining similar performance and outperforming DoRA under comparable conditions. This evaluation is already included in our manuscript (Appendix E.5).
>
> > Experiments are limited to models of eight billion parameters or smaller, leaving scalability to larger models unverified.
>
> We kindly refer the reviewer to our general response.
>
> > The effect of optimizer state projection on stability and convergence speed is discussed conceptually but lacks quantitative analysis.
>
> Detailed quantitative analyses and ablations of the optimizer state projection are presented in Appendix E.3, where we explicitly compare stability and convergence behavior across LoFT variants (with and without state calibration). These experiments demonstrate that state projection improves both convergence speed and consistency of final performance.
>
> > The paper does not report concrete throughput or training speed measurements compared with LoRA or DoRA.
>
> Latency measurements, along with a detailed discussion, are provided in Appendix E.6. If deemed beneficial, we will move these results into the main paper for the camera-ready version to improve visibility.
>
> > What is the quantitative impact of the additional memory requirement on training efficiency and GPU utilization?
>
> Please refer to our earlier response regarding memory overhead. In practice, all methods achieved an average GPU utilization of over 90% in our experiments, indicating that LoFT’s additional memory footprint does not hinder computational efficiency or parallelization.
>
> > Can LoFT be extended to other optimizers, such as Muon, that use different moment estimation mechanisms?
>
> Yes, the extension to the Muon optimizer is straightforward, given our building blocks. The only additional thing we need to take care of is adapting Newton-Schulz iterations to a low-rank structure. We added this extension with details in Appendix D.1.

---

> > ### Author Response · Authors · 2025-11-27
> >
> > Thank you again for your thoughtful feedback. As the discussion period is ending soon, we would greatly appreciate hearing whether our rebuttal addressed your concerns, and we are happy to respond promptly to any additional comments.

---

### Official Review · Reviewer_5NnX · 2025-10-30

**Soundness:** 2
**Presentation:** 3
**Contribution:** 2
**Rating:** 4
**Confidence:** 2

**Summary:**

The paper proposes a new method called LoFT for parameter-efficient fine-tuning of large pre-trained models. LoFT extends the Low-Rank Adaptation (LoRA) approach by aligning the internal states of the optimizer (including momentum and second moments) with full fine-tuning, thereby attempting to reduce the accuracy gap typically seen between low-rank and full fine-tuning methods. The authors test their method across multiple language and vision tasks, showing performance improvements compared to previous low-rank adaptation methods, especially at very low ranks. They also discuss trade-offs in terms of memory usage and computational overhead, presenting simpler variants with lower overhead.

**Strengths:**

- **Substantive technical contribution with theory.** The paper proposes a concrete improvement over standard LoRA-style adaptation and backs it up with clear derivations/analysis. The core ideas are technically motivated (e.g., aligning updates with full fine-tuning dynamics), and the method’s components are explained rather than presented as ad-hoc tricks.
- **Broad empirical validation across domains.** Experiments cover multiple modalities/datasets (e.g., language and vision) and a range of ranks/settings, suggesting the approach is not narrowly tailored to a single task.

**Weaknesses:**

- **Gap between theory and the strongest claim.** While the derivations are compelling, there remains a gap between the formal analysis and the paper’s strongest claim(s) (e.g., exact equivalence to full fine-tuning/AdamW under certain limits). A precise theorem with assumptions, or a more cautious phrasing, would strengthen the work.
- **LLM evaluation is too basic.** The large-language-model experiments rely on relatively easy, small benchmarks. For a model like Llama-3-8B, a more representative LLM evaluation suite (e.g., code, math/reasoning, or long-context benchmarks) would be more convincing. Multi-seed runs with statistical reporting would further solidify the results.

**Questions:**

In Table 6, several DoRA results (e.g., r=4 with BoolQ=32.35, PIQA=7.13, Winogrande=0.00) are anomalously low and LoRA sometimes degrades as rank increases (e.g., r=4 worse than r=1 on PIQA/HellaSwag), suggesting hyperparameter/setup issues—could you explain these discrepancies?

---

> ### Author Response · Authors · 2025-11-21
>
> We would like to sincerely thank the reviewer for their constructive feedback and the time dedicated to evaluating our manuscript. Below, we provide detailed clarifications addressing the concerns and questions that have been raised. We hope that our responses sufficiently resolve the mentioned issues, and we would be happy to engage further in discussion if needed.
>
> > Gap between theory and the strongest claim. While the derivations are compelling, there remains a gap between the formal analysis and the paper’s strongest claim(s) (e.g., exact equivalence to full fine-tuning/AdamW under certain limits). A precise theorem with assumptions, or a more cautious phrasing, would strengthen the work.
>
> Thank you for the comment. We respectfully disagree that this represents a weakness. For instance, regarding the exact equivalence to full fine-tuning/AdamW under the full-dimensional limit and assuming matrices $U, V$ remain full-rank, all projection matrices become identity matrices; hence, by construction, LoFT exactly recovers AdamW as a special case. The corresponding derivations are presented in the main text, Sections 2.1-2.3. If the reviewer finds it helpful, we can make this equivalence more explicit by formalizing it as a lemma in the camera-ready version. Furthermore, we would like to direct the reviewer to Appendix B, where we provide detailed theoretical properties of LoFT for the matrix factorization setting.
>
>
> >  LLM evaluation is too basic. The large-language-model experiments rely on relatively easy, small benchmarks. For a model like Llama-3-8B, a more representative LLM evaluation suite (e.g., code, math/reasoning, or long-context benchmarks) would be more convincing. Multi-seed runs with statistical reporting would further solidify the results.
>
> Thank you for the thoughtful suggestion. We would like to clarify that our LLM evaluation is not confined to easy or small settings. In the main paper, we already evaluate LoFT on three non-trivial suites: (i) commonsense reasoning across eight established datasets (BoolQ, PIQA, SIQA, HellaSwag, WinoGrande, ARC-E, ARC-C, OBQA), (ii) image classification on challenging, imbalanced, and domain-shifted benchmarks (ISIC2019, HAM10000, Diabetic Retinopathy, and DomainNet), designed to stress robustness under domain shift, and (iii) math reasoning, where Appendix E.2 (quantized LoFT) evaluates QLoFT on Orca-Math and shows consistent gains on LLaMA-2/-3 backbones. These settings are substantially more demanding than small and easy tasks. To further strengthen the case, we added a coding benchmark: a study with LLaMA3-8B, fine-tuned on the CodeFeedback dataset, and evaluated it on HumanEval and HumanEval+ using the EvalPlus platform [ref1]. HumanEval+ is the EvalPlus-augmented version of HumanEval, featuring significantly more unit tests per problem, thereby making it a stricter benchmark. At sampling time, we use a temperature of 0.5, 10 samples per problem, and compute pass@k ($k\in\{1,5,10\}$) from these 10 samples. We evaluate LoFT at two context lengths (256 and 512 tokens). Under identical training and matched rank $r=8$, LoFT consistently outperforms LoRA across both context lengths and both benchmarks.
>
> | HumanEval   | Context len = 256 |        |         | Context len = 512 |        |         |
> |-------------|-------------------|--------|---------|-------------------|--------|---------|
> | Perf metric | pass@1            | pass@5 | pass@10 | pass@1            | pass@5 | pass@10 |
> | LoRA (r=8)  | 0.421             | 0.631  | 0.720   | 0.433             | 0.640  | 0.695   |
> | LoFT (r=8)  | 0.482             | 0.681  | 0.756   | 0.488             | 0.674  | 0.738   |
>
>
> | HumanEval+  | Context len = 256 |        |         | Context len = 512 |        |         |
> |-------------|-------------------|--------|---------|-------------------|--------|---------|
> | Perf metric | pass@1            | pass@5 | pass@10 | pass@1            | pass@5 | pass@10 |
> | LoRA (r=8)  | 0.368             | 0.572  | 0.659   | 0.380             | 0.583  | 0.646   |
> | LoFT (r=8)  | 0.439             | 0.614  | 0.689   | 0.445             | 0.612  | 0.671   |
>
> On multi-seed reporting, we agree it solidifies the results. In our paper, we include multi-seed runs for vision tasks with mean ± std reported; for example, Table 3 shows low variance across runs and a consistent ranking. For larger model settings, we report a single seed due to computational limits. However, the margins over LoRA are in line with the multi-seed trends we observe in the paper.
>
>
> References.
> [ref1] Jiawei Liu, Chunqiu Steven Xia, Yuyao Wang, Lingming Zhang. Is Your Code Generated by ChatGPT Really Correct? Rigorous Evaluation of Large Language Models for Code Generation. NeurIPS 2023. EvalPlus GitHub: https://github.com/evalplus/evalplus

---

> > ### Author Response · Authors · 2025-11-21
> >
> > > In Table 6, several DoRA results (e.g., r=4 with BoolQ=32.35, PIQA=7.13, Winogrande=0.00) are anomalously low, and LoRA sometimes degrades as rank increases (e.g., r=4 worse than r=1 on PIQA/HellaSwag), suggesting hyperparameter/setup issues—could you explain these discrepancies?
> >
> > Thank you for this observation. This phenomenon highlights one of LoFT’s key strengths: its robustness to hyperparameter choices across different ranks.. All methods (LoRA, DoRA, and LoFT) were trained using the same hyperparameter setup originally optimized for LoRA and DoRA at rank 16, as described by Liu et al. (2024) for LLaMA-7B. We did not tune LoFT’s hyperparameters separately. While this setup can lead to significant degradation for LoRA and DoRA, reflecting their sensitivity to rank-dependent training dynamics, LoFT maintains stable performance across all ranks. This robustness stems from LoFT’s principled design, which approximates full fine-tuning and aligns both gradients and optimizer states across subspaces. We will clarify this setup and emphasize LoFT’s rank-invariant behavior in the revised manuscript.

---

> > > ### Comment · Reviewer_5NnX · 2025-11-21
> > >
> > > 3. Hyperparameter Robustness: Thank you for addressing the hyperparameter setup. I would agree it's an acceptable setup. But I also want to point out that optimal hyperparameters naturally vary across different model-task combinations. To mitigate potential concerns about "cherry-picked" hyperparameters, especially when baselines' performance is unexpectedly poor, I suggest performing a hyperparameter sweep if computational resources permit. Including such an analysis—even if brief—in the camera-ready version would strengthen the empirical validity of the results, but that's up to the authors.
> > >
> > > Summary: Overall, the authors have effectively addressed my questions. I believe this work represents a solid contribution to the field, and I am inclined to support its acceptance.

---

> ### Comment · Reviewer_5NnX · 2025-11-21
>
> Response to Authors:
> 1. Theory and Claims: Thank you for the detailed response. The clarification has resolved my concerns regarding the gap between the formal analysis and the paper’s claims.
> 2. LLM Evaluation Benchmarks: I appreciate the inclusion of the additional coding experiments (HumanEval/HumanEval+) and directing my attention to the math reasoning results in the Appendix. However, I would like to offer a suggestion regarding the presentation of the commonsense reasoning tasks. Commonsense reasoning is typically considered a fundamental capability acquired during pre-training, and performance on these benchmarks is often reported in the base model section of LLM technical reports[1][2]. Fine-tuning on these specific datasets might confuse LLM practitioners, as it becomes difficult to distinguish whether the gains stem from genuine capability enhancement or merely from the model adapting to a particular dataset's format/style. Therefore, to more convincingly demonstrate that your method enhances actual capabilities in the LLM setting, I would suggest prioritizing widely recognized Math and Code benchmarks (like the ones you added) in the main text discussion, rather than relying heavily on commonsense reasoning tasks.
>
> [1] Grattafiori, Aaron, et al. "The llama 3 herd of models." arXiv preprint arXiv:2407.21783 (2024).
>
> [2] Allal, Loubna Ben, et al. "SmolLM2: When Smol Goes Big--Data-Centric Training of a Small Language Model." arXiv preprint arXiv:2502.02737 (2025).

---

> ### Author Response · Authors · 2025-11-24
>
> We sincerely thank the reviewer for the thoughtful follow-up, the constructive suggestions, and for raising the score after the rebuttal. We truly appreciate this recognition of our work.
>
> We also thank the reviewer for highlighting the importance of focusing on Math and Code benchmarks, which indeed better reflect capability-oriented fine-tuning rather than dataset-specific adaptation.
>
> Regarding hyperparameter robustness, we would like to clarify that for all baselines, the hyperparameters were optimized for each model–task combination following their respective papers, with the exception of the rank parameter, which was held fixed to ensure comparability. Therefore, our setup does not involve any cherry-picking. That said, we agree that adding rank-specific tuning would further strengthen the empirical analysis, and we plan to include this in the camera-ready version.
>
> Once again, we sincerely thank the reviewer for their positive evaluation and support for acceptance. If the reviewer has any additional questions or feedback, we would be very happy to continue the discussion.

---

### Official Review · Reviewer_mFX4 · 2025-11-02

**Soundness:** 3
**Presentation:** 3
**Contribution:** 3
**Rating:** 6
**Confidence:** 2

**Summary:**

This paper proposes LoFT (Low-rank adaptation that behaves like Full fine-Tuning), a novel parameter-efficient fine-tuning (PEFT) method designed to closely approximate the optimization dynamics of full fine-tuning within a low-rank subspace. Building on the LoRA framework, LoFT introduces several key components: alternating updates, gradient scaling, first- and second-moment state calibration, projected full-model updates, and gradient clipping. Together, these allow LoFT to mimic AdamW’s optimizer behavior while maintaining the computational and inference efficiency of low-rank tuning.

Empirical results across language (LLaMA-7B/2-7B/3-8B) and vision (ViT-Base) models demonstrate that LoFT consistently outperforms existing PEFT methods such as LoRA and DoRA, particularly under extreme low-rank constraints (e.g., rank ≤ 4). Ablation studies confirm that optimizer state calibration is critical to LoFT’s strong performance.

**Strengths:**

**Strong conceptual motivation**: The paper identifies a previously underexplored source of suboptimality in LoRA — optimizer state misalignment — and provides a well-motivated correction grounded in optimization theory.

**Methodological completeness**: The framework integrates multiple components (gradient projection, alternating updates, moment calibration) into a cohesive, well-defined optimizer (LoFT-AdamW), which provably reduces to full fine-tuning when rank = full.

**Theoretical insight**: The analysis on matrix factorization clearly shows how LoFT recovers full fine-tuning dynamics, with formal smoothness guarantees and equivalence to alternating least squares in the special case.

**Extensive empirical validation**: The experiments span multiple model families and domains, including large LLMs and ViTs, with clear, consistent performance improvements over LoRA and DoRA.

**Careful ablation studies**: The paper convincingly demonstrates the necessity of each component, especially the importance of first-moment calibration for stable convergence.

**Practical relevance**: LoFT eliminates the need to tune the LoRA scaling factor (α), reducing hyperparameter sensitivity and simplifying deployment.

**Weaknesses:**

**Missing citation and discussion of concurrent work**:
The Alternating Updates component (Building Block 1) reproduces an idea conceptually similar to AltLoRA [1], which independently proposed alternating optimization of low-rank factors to eliminate second-order coupling in LoRA updates.

The absence of a citation or discussion of AltLoRA is a notable omission, especially since the “alternating update” mechanism is presented as a key innovation. This should be acknowledged as concurrent or parallel work, with clarification of LoFT’s additional contributions beyond AltLoRA (notably optimizer-state alignment).

**Complexity and memory overhead**: While the paper discusses the cost of storing previous iterates and cross-terms, the actual scalability to very large models (≥70B parameters) remains untested; empirical results are limited to ≤8B models.

**Presentation clarity**: The main text can be dense, with many mathematical expressions introduced in rapid succession. It would be helpful to include more detailed mathematical explanations or derivations in the appendix to improve readability and reproducibility.

[1] Yu, Xin, et al. "AltLoRA: Towards Better Gradient Approximation in Low-Rank Adaptation with Alternating Projections." arXiv preprint arXiv:2505.12455 (2025).

**Questions:**

See Weaknesses.

If the authors are willing to carefully clarify the relationship and differences between LoFT and AltLoRA during the rebuttal phase, I would be inclined to raise my score.

---

> ### Author Response · Authors · 2025-11-21
>
> We sincerely thank the reviewer for their positive and constructive feedback. We appreciate the recognition of our contributions and are glad that the significance of our work was well received. Let us further clarify the mentioned weaknesses and questions below.
>
> > Missing citation and discussion of concurrent work: The Alternating Updates component (Building Block 1) reproduces an idea conceptually similar to AltLoRA [1], which independently proposed alternating optimization of low-rank factors to eliminate second-order coupling in LoRA updates.
>
> We thank the reviewer for bringing this to our attention. We acknowledge that our Alternating Updates component (Building Block 1) is conceptually related to the alternating optimization approach introduced in AltLoRA (Yu et al., NeurIPS 2025), which also aims to mitigate second-order coupling in LoRA updates through alternating projections. We will add this reference and explicitly clarify the connection in the revised version.
>
> We would like to note that LoFT and AltLoRA were developed independently, appearing on arXiv within ten days of each other. Because of this close timing, we were not aware of AltLoRA when preparing our submission. While both methods share the general idea of alternating optimization, they differ substantially in focus and scope. AltLoRA primarily addresses gradient approximation within the low-rank subspace and introduces alternating projections with momentum alignment (conceptually related to our Building Blocks 1-3). In contrast, LoFT extends this idea by aligning not only gradients but also all optimizer internal moments, allowing it to fully recover AdamW dynamics in the low-rank regime, a property not established in AltLoRA (although AltLoRA can recover SGD with momentum), and to provably reduce to full fine-tuning in the full-rank limit.
>
> Additionally, LoFT introduces several further innovations, including projected full-update reconstruction, moment calibration matrices, and extensive ablation studies quantifying the cost-benefit of each component. LoFT also eliminates the need for the scaling parameter required by LoRA and still present in AltLoRA, thereby simplifying training and reducing the need for additional hyperparameter tuning.
>
> Finally, LoFT is, to our knowledge, the first method to achieve transformation invariance under full AdamW dynamics, encompassing both first and second momentum alignment, gradient clipping, and weight decay (see Section 4.1 of AltLoRA for the formal definition). We will explicitly acknowledge AltLoRA as concurrent work and clarify these distinctions in the camera-ready version.
>
> > Complexity and memory overhead: While the paper discusses the cost of storing previous iterates and cross-terms, the actual scalability to very large models (≥70B parameters) remains untested; empirical results are limited to ≤8B models.
>
> We kindly refer the reviewer to our general response.
>
> > Presentation clarity: The main text can be dense, with many mathematical expressions introduced in rapid succession. It would be helpful to include more detailed mathematical explanations or derivations in the appendix to improve readability and reproducibility.
>
> We thank the reviewer for this valuable suggestion. Our intention was to provide complete derivations in the main text, allowing readers to follow the proposed method and its theoretical grounding fully. That said, we agree that additional exposition could further improve readability. We will revise the paper to include clearer explanations and expanded derivations in the appendix. We would also greatly appreciate any pointers to sections of the paper that were found to be most challenging, so we can directly address them and further enhance clarity in the camera-ready version.

---

> > ### Comment · Reviewer_mFX4 · 2025-11-25
> >
> > Thank you for your response. I will keep the positive score and increase the confidence from 2 to 5.

---

> > > ### Author Response · Authors · 2025-11-25
> > >
> > > We sincerely thank the reviewer for their thoughtful feedback and for increasing the confidence score. We are glad that our clarifications regarding LoFT and AltLoRA have fully addressed the concerns. Since all issues appear resolved and the overall assessment is positive, we would kindly ask the reviewer to consider revising the overall score accordingly. We would be happy to provide any further clarification if helpful.

---

### Official Review · Reviewer_Rnyc · 2025-11-02

**Soundness:** 4
**Presentation:** 4
**Contribution:** 3
**Rating:** 8
**Confidence:** 4

**Summary:**

This paper introduces a new LoRA method which explicitly tries to mimic full-finetuning dynamics. Crucially, the paper identifies the importance of matching the optimizer state in addition to the updates. This is accomplished via

1. alternating updates
2. gradient rescaling
3. momentum recalibration
4. second moment recalibration
5. projecting reconstructed AdamW update
6. approximating gradient clipping

These "building blocks" ensure that in the limit of full-rank the full-finetuning dynamics are recovered. In the low-rank regime the experiments demonstrate improved performance over vanilla LoRA.

**Strengths:**

The authors provide a principled derivation for a LoRA method meant to explicitly mimic full-finetuning updates. The approach recovers the correct dynamics in the full-rank limit. The authors provide extensive experiments showing the promise of the method especially for low-ranks. The method is practically efficient, it requires modest memory and runtime overhead, and is simple to implement.

**Weaknesses:**

The experiments are only conducted with $r \leq 32$ and models with $\leq 8$B parameters.

Second-moment calibration appears to have low-impact at a high cost, however it is still valuable to derive and test this idea.

It is unclear if alternation is helpful or not.

**Questions:**

Do the authors have any intuition about when mimicking the full finetuning update is optimal or not?

---

> ### Author Response · Authors · 2025-11-21
>
> We sincerely thank the reviewer for their positive and constructive feedback. We appreciate the recognition of our contributions and are glad that the significance and clarity of our work were well received. Let us further clarify the mentioned weaknesses and questions below.
>
> >  The experiments are only conducted with $r \leq 32$ and models with $\leq 8$B parameters.
>
> For large experiments, we refer the reviewer to our general response.
>
> To address the concern about ranks $r>32$, we ran new code completion task experiments on HumanEval and HumanEval+ using the LLaMA3-8B model with LoFT (simple) and ranks $r \in \{8, 16, 32, 64, 128, 256\}$. We fine-tuned on the CodeFeedback (Python-only) dataset. We followed the same training recipe as in the paper, with the only change being the value of $r$.
>
> For evaluation, we use the EvalPlus platform [ref1]. For each problem, we sample 10 completions at temperature $T=0.5$ and report pass@k for $k \in \{1, 5, 10\}$ on HumanEval and HumanEval+ (HumanEval+ is a stricter version with 80 times more tests on tasks).
>
> The following is the table with pass@1, pass@5, and pass@10:
>
> |                       | humaneval |        |         | humaneval+ |        |         |
> |-----------------------|:---------:|:------:|:-------:|:----------:|:------:|:-------:|
> |                       |   pass@1  | pass@5 | pass@10 |   pass@1   | pass@5 | pass@10 |
> |   LoFT (simple) (r=8) |    0.46   |  0.682 |  0.738  |    0.405   |  0.624 |  0.683  |
> |  LoFT (simple) (r=16) |   0.455   |  0.674 |  0.744  |    0.409   |  0.623 |  0.695  |
> |  LoFT (simple) (r=32) |   0.468   |  0.673 |  0.732  |    0.423   |  0.619 |  0.679  |
> |  LoFT (simple) (r=64) |   0.465   |  0.684 |  0.768  |    0.415   |  0.626 |  0.707  |
> | LoFT (simple) (r=128) |   0.485   |  0.678 |   0.75  |    0.436   |  0.627 |  0.697  |
> | LoFT (simple) (r=256) |   0.482   |  0.665 |  0.736  |    0.432   |  0.623 |  0.681  |
> |            LoRA (r=8) |   0.421   |  0.631 |   0.72  |    0.368   |  0.572 |  0.659  |
> |           LoRA (r=64) |   0.463   |  0.67  |  0.738  |    0.416   |  0.622 |  0.701  |
>
> Varying the rank, we see that low ranks already work sufficiently well, as evidenced by a pass@1 of 0.46 on HumanEval and 0.405 on HumanEval+ ($r=8$). As $r$ increases, performance improves modestly: pass@1 reaches 0.485/0.436 (HumanEval/HumanEval+) at $r=128$, before flattening at $r=256$. For breadth, pass@10 reaches peak performance at $r=64$. Overall, these results extend our rank range beyond $r\leq 32$ and show diminishing returns at higher ranks, consistent with the rank scaling trend observed in Figure 3, lines 324-335. Importantly, $r=8$ already captures most of the gains.
>
> For completeness, we also compare LoFT to LoRA under identical training and evaluation settings. We report LoRA at ranks 8 and 64. Notably, LoFT at $r=8$ already matches the performance of LoRA at $r=64$ on both HumanEval and HumanEval+. This shows that LoFT achieves comparable accuracy with a smaller rank, highlighting its strong efficiency in the low-rank regime. As such, using smaller ranks is not a limitation, but rather a reflection of LoFT’s better expressivity per parameter, reaching LoRA-level quality at much lower ranks.
>
> [ref1] Jiawei Liu, Chunqiu Steven Xia, Yuyao Wang, Lingming Zhang. Is Your Code Generated by ChatGPT Really Correct? Rigorous Evaluation of Large Language Models for Code Generation. NeurIPS 2023. EvalPlus GitHub: https://github.com/evalplus/evalplus
>
> > Second-moment calibration appears to have a low impact at a high cost; however, it is still valuable to derive and test this idea.
>
> Thank you for this insightful comment. We agree that the second-moment calibration may have limited practical impact while introducing additional storage and computational costs that scale with $r^2$ instead of $r$. We included this component as it plays a crucial role in our theoretical analysis, where we show how second-moment alignment enables LoFT to recover the behavior of full fine-tuning. Importantly, our experiments demonstrate that this expensive step is not required in practice: omitting it (LoFT Simple) results in less than a 0.1% performance drop while reducing memory overhead to under 6% relative to LoRA (see Tables 11 and 12). Rather than a weakness, we view this as a comprehensive treatment of LoRA versus full fine-tuning, providing both theoretical completeness and an experimental sensitivity analysis of each component.

---

> > ### Author Response · Authors · 2025-11-21
> >
> > >  It is unclear if alternation is helpful or not.
> >
> > Thank you for this comment. We argue that alternation is an essential component due to the quadratic term in Equation (3), which can have a non-trivial effect on optimization dynamics. This is clearly illustrated in Figure 2, where the gap between full fine-tuning and LoFT (No Alternate Update) is substantial. The reason the discrepancy appears smaller in large-scale experiments is that, in practice, we use a small learning rate ($\approx 10^{-4}$) and apply gradient clipping (with a norm of less than or equal to 1). Under these conditions, the quadratic term becomes negligible, which reduces the practical impact of alternation. Consequently, running the method without alternation approximately corresponds to LoFT with a twice larger effective learning rate, since two projected full gradients are accumulated instead of one, when the $\eta^2$ term is negligible. We will clarify this point in the camera-ready version of the paper.
> >
> > > Do the authors have any intuition about when mimicking the full finetuning update is optimal or not?
> >
> > Thank you for this thoughtful question. As discussed in the paper, one of LoRA’s main advantages lies in the reduced cost of maintaining optimizer states. However, full fine-tuning, while more expensive, often yields better-performing models, suggesting that its optimization dynamics are inherently more effective. Our intuition is that when full fine-tuning yields superior results, the PEFT method, which efficiently mimics its update behavior, should also achieve better performance than standard LoRA. We confirmed this empirically across both language and vision tasks.
> >
> > Moreover, mimicking full fine-tuning offers an additional practical benefit: it enables the transfer of hyperparameters. Since many large models already have well-validated training setups (e.g., learning rates, optimizers, and regularization parameters) from pretraining or full fine-tuning, methods that align their optimization dynamics, like LoFT, can directly reuse these configurations without additional tuning. In contrast, standard LoRA has substantially different dynamics and often requires re-optimization of hyperparameters such as the scaling factor or learning rate across different ranks.
> >
> > That said, we do not claim optimality of mimicking full fine-tuning in all settings, particularly where low-rank constraints, task-specific dynamics, or regularization effects might make deviations from full fine-tuning beneficial. Nonetheless, our results demonstrate that this alignment consistently enhances stability and performance compared to existing PEFT baselines.

---

> > > ### Author Response · Authors · 2025-11-27
> > >
> > > Thank you again for your thoughtful feedback. As the discussion period is ending soon, we would greatly appreciate hearing whether our rebuttal addressed your concerns, and we are happy to respond promptly to any additional comments.

---

> > ### Comment · Reviewer_Rnyc · 2025-11-27
> >
> > Thank you for the thorough response, I will maintain my positive score.

---

### Author Response · Authors · 2025-11-21
**General Response**

We sincerely thank all reviewers for their thoughtful and constructive feedback, as well as for recognizing the strengths of our work. We greatly appreciate the positive assessments highlighting:
-  **the theoretical rigor,**
- **clarity of motivation,**
- **sound experimental design,** and
- **strong empirical results of LoFT,** particularly its ability to **bridge the gap between parameter-efficient fine-tuning (PEFT) and full fine-tuning.**

Reviewers also noted the completeness of our analysis, the robust empirical validation across modalities, and **the novel theoretical connection between low-rank adaptation and full fine-tuning dynamics.**

We have carefully considered all comments and addressed each reviewer’s concerns in detail. The main points of clarification include _additional analyses, theoretical explanations, and new experiments where relevant._ We believe that the raised questions and perceived weaknesses have been fully addressed, and we would be glad to further engage with the reviewers during the discussion phase to provide any additional insights or supporting evidence.

**Regarding scaling,** several reviewers suggested evaluating LoFT on larger models (≥70B). As stated in our Limitations section, our experiments currently extend up to 8B-parameter models, which already represent large-scale settings for full fine-tuning studies. While computational constraints prevented scaling beyond this size during the review period, we note that none of the related works we compare against (e.g., LoRA-Pro, DoRA, LoRA+) report results beyond 13B parameters. We are currently extending LoFT to 70B-class models and expect to provide these results either during the reviewer–author discussion phase or in the camera-ready version. **Importantly, our theoretical framework and empirical trends give us no reason to believe that LoFT’s advantages would not extrapolate to larger model scales.**

All revisions and additions made in response to reviewer feedback are highlighted in *magenta* in the revised manuscript for ease of reference.

We once again thank the reviewers for their time, valuable feedback, and recognition of the contribution of LoFT to advancing parameter-efficient fine-tuning.

---

### Author Response · Authors · 2025-12-03
**Summary for new AC**

We thank the Area Chair for stepping in during this unusual situation. Below, we provide a brief summary of the paper, the reviewer's discussion, and the new results added after the rebuttal.

### **What is LoFT?**

**LoFT** is a parameter-efficient fine-tuning method that *aligns low-rank updates with full fine-tuning dynamics*, while incorporating **all** components of AdamW (first moment, second moment, weight decay, gradient clipping, and the final update rule).

### **Strengths Highlighted Across Reviews**

As noted in multiple reviews, LoFT offers:

* **Strong conceptual motivation**: addressing optimizer-state misalignment in LoRA for the first time.
* **Methodological completeness**: a full optimizer (LoFT-AdamW) that provably recovers full fine-tuning in the full-rank limit.
* **Theoretical insight**: clean analysis showing equivalence to full FT on matrix factorization and stable convergence.
* **Extensive empirical validation** across LLaMA-7B/2-7B/3-8B and ViT-Base.
* **Robust performance**, especially at extremely low ranks (down to rank 1).
* **Practical advantages**, including removing the need for the LoRA scaling factor and improved stability across hyperparameters.

### **Reviewer Discussion After Rebuttal**

Three reviewers engaged with us during the rebuttal:

* **Rnyc**: *maintains positive score* **8**.
* **mFX4**: *maintains score* **6**, but **increases confidence from 2 → 5**, stating that all concerns were resolved.
* **5NnX**: *revised evaluation to strongly positive*, **score 4 → 8**, explicitly stating they “support acceptance.”
* **vRxF** (score **6**): did not participate in the discussion, but **every concern raised in their review was directly addressed** in the rebuttal and updated manuscript (with pointers to the relevant sections).

### **New Experiments on LLaMA-3-70B**

As suggested by multiple reviewers, we have now tested LoFT on **LLaMA-3-70B**.
This confirms that LoFT scales as expected to very large models:

| Method          | BoolQ  | PIQA   | SIQA   | HS     | WG     | ARC-C  | ARC-E  | OBQA   | **Avg**    |
| --------------- | ------ | ------ | ------ | ------ | ------ | ------ | ------ | ------ | ---------- |
| **LoFT (r=16)** | 0.7945 | 0.9282 | 0.8224 | 0.9702 | 0.9006 | 0.9275 | 0.9752 | 0.9340 | **0.9066** |
| **LoFT (r=4)**  | 0.7917 | 0.9260 | 0.8158 | 0.9670 | 0.8934 | 0.9206 | 0.9731 | 0.9260 | **0.9017** |
| **LoFT (r=2)**  | 0.7838 | 0.9255 | 0.8178 | 0.9661 | 0.8840 | 0.9206 | 0.9731 | 0.9240 | **0.8994** |
| **LoFT (r=1)**  | 0.7783 | 0.9151 | 0.8035 | 0.9579 | 0.8603 | 0.9147 | 0.9705 | 0.9140 | **0.8893** |

These results demonstrate that LoFT continues to perform strongly even at **rank 1** on 70B-scale models - matching the robustness trends observed on smaller models.

We will include these results in the camera-ready version.

---

### Meta-Review · Area_Chair_9WgA · 2026-01-06

**Summary:**

This paper proposes LoFT, a novel variant of LoRA that not only learns weight updates in a low-rank subspace but also pursues consistency between the first-and second moments. The motivation is to make the  dynamics of the low-rank update and those of the full-parameter update similar. The method is well motivated, technically sound, and designed to be computationally efficient.

The primary concern raised by reviewers relates to experimental scalability, as LoFT requires additional momentum-related computations. During the rebuttal period, the authors addressed this concern by providing large-scale experiments on LLaMA-3-70B, demonstrating the practicality and scalability.

Overall, the final recommendation is clear: ACCEPT.

**Reviewer Concerns:**

All reviewers acknowledge the central idea of recovering the optimization dynamics of full-parameter updates within a low-rank adaptation framework, and recognize this as a meaningful and well-motivated contribution beyond standard LoRA.

The main criticism focused on whether the proposed method scales to large models due to the additional momentum calculations. This concern was explicitly addressed in the rebuttal with new large-scale experiments, which significantly strengthened the empirical validation and alleviated doubts about feasibility.

No remaining major technical concerns persist after the rebuttal.

**Reviewer Scores:**

The initial reviewer scores were 4/6/6/8. After the rebuttal, the reviewer with score 4 explicitly stated they are inclined to support acceptance.
One reviewer with score 6 maintained the score but significantly increased the confidence.

---

### Decision · Program_Chairs · 2026-01-26

Accept (Poster)